# Subspace-Aware Feature Reshaping for Open-Set Graph Class-Incremental Learning

**Weichao Zhang** [1]  **Shuai Zheng** [1 2]  **Yeyu Yan** [1 2]  **Zhizhe Liu** [3]  **Zhenfeng Zhu** [1 2]  **Yao Zhao** [1 2]

## Abstract

Graph class-incremental learning (GCIL) has emerged to address the challenge of learning from dynamically evolving graphs, which continuously learns new classes over a sequence of tasks while retaining performance on previously seen classes. However, existing GCIL methods assume a closed-set test distribution drawn only from seen tasks. This fundamentally contradicts real-world open-ended scenarios where future unknown classes inevitably emerge. Empirically, we observe that existing GCIL methods falter in such open-set settings due to severe representation drift and generalized overconfidence. To bridge this gap, we investigate the Open-Set GCIL problem and propose **SAFER** (Subspace-Aware FEature Reshaping), a novel framework that endows GCIL with intrinsic open-set capabilities under a replay-free constraint. Specifically, **SAFER** performs subspace-aware feature reshaping with drift-resilient fingerprints, unifying task routing and open-set rejection into a single energy-based metric. Furthermore, we introduce a geometric space-consistency regularization that explicitly improves intra-class compactness and suppresses cross-task representation drift. Extensive experiments on four benchmarks demonstrate that SAFER outperforms state-of-the-art baselines by margins of up to 5.2% in accuracy and 31.3% in open-set AUROC, all while maintaining near-zero forgetting under strict no-replay constraints. The code has been released in https://github.com/ZhangWeichao0824/SAFER

## 1. Introduction

Graphs are widely used to model relational data in applications such as social networks (Zhang et al., 2022c; Chen et al., 2026), citation networks (Chen et al., 2025) and recommender systems (Anand & Maurya, 2025). GNNs (Kipf, 2016; Veličković et al., 2017; Xu et al., 2018; Wu et al., 2019; Zheng et al., 2025) have become a standard approach, which learn expressive node(Zheng et al., 2020; 2023) by aggregating structural and semantic information, supporting downstream tasks such as node classification (Xiao et al., 2022) and link prediction (Zhang & Chen, 2018). In many real deployments, graphs are highly dynamic, which requires models to be updated incrementally without expensive retraining, while avoiding catastrophic forgetting of previously learned knowledge. Continual graph learning (CGL) (Febrinanto et al., 2023; Tian et al., 2024; Zhang et al., 2024b) has been proposed to address this setting by learning from a sequence of tasks while preserving performance on past ones.

Despite substantial progress in mitigating forgetting (Zhou & Cao, 2021; Liu et al., 2021; Zhang et al., 2022b), existing CGL methods typically rely on a rigid closed-world assumption, i.e., they presume test samples are drawn exclusively from known classes observed during training. While this closed-set assumption simplifies evaluation, it fundamentally contradicts real-world open-ended deployments, where the graph topology is continuously expanding and nodes from future, unseen tasks inevitably emerge in the data stream. Under this constraint, classifiers lack the option to reject anomalies and are forced to project out-of-distribution inputs onto the nearest historical decision boundaries. Consequently, models suffer from generalized overconfidence, erroneously assigning high probabilities to unknown samples and accumulating errors over time.

To demystify the underlying failure mechanism, our analysis dissects a coherent causal chain inherent in standard GCIL. Symptomatically, we observe that unknown samples indiscriminately elicit high confidence scores comparable to known classes, rendering simple threshold-based rejection ineffective. We trace this phenomenon to its root cause: Catastrophic Representation Drift. As the graph encoder continuously adapts to the topology of new tasks, the ge-

[1]School of Computer Science & Technology , Beijing Jiaotong University, Beijing, China [2]Visual Intelligence + X International Cooperation Joint Laboratory of the Ministry of Education, China [3]School of Information and Communication Engineering, Beijing Information Science and Technology University, Beijing, China. Correspondence to: Shuai Zheng <ShuaiZheng@bjtu.edu.cn>.

*Proceedings of the 43rd International Conference on Machine Learning*, Seoul, South Korea. PMLR 306, 2026. Copyright 2026 by the author(s).

| Task 0 | Tasks 0-1 | Tasks 0-2 | Tasks 0-3 | Tasks 0-4 |

*Figure 1.* t-SNE evolution of representations across tasks. As tasks progress, seen tasks (colored) form new clusters while largely preserving earlier cluster geometry, whereas unseen tasks (gray) are treated as unknown and remain separated from the known regions.

ometric manifolds of historical classes are progressively eroded and displaced. This structural distortion dismantles the stable metric reference required to delineate the boundary between known concepts and the open world.

Motivated by the observations, we propose a new framework called **SAFER** (Subspace-Aware FEature Reshaping) that, under a no-replay constraint, simultaneously addresses cross-task forgetting and inference-time detection of unknown samples. Considering the low-rank nature of graph spectral features, we seek a cross-task comparable criterion that remains stable as the task stream on the graph evolves.

To achieve this, SAFER constructs drift-resilient task subspace fingerprints to explicitly characterize the low-rank geometric manifold of each seen task. Leveraging these fingerprints as geometric bases, we dynamically reshape the query representation by projecting it onto the subspace of each task. The geometric cost of this reshaping, quantified as the reconstruction residual, serves as a unified energy score that selects the most compatible task while reliably rejecting unknown samples. Meanwhile, we introduce a space-consistency regularization enhanced by structural priors to stabilize within-task geometry, suppress intra-class expansion, and preserve inter-class separability without replay, thereby keeping task fingerprints identifiable over time. As illustrated in Figure 1, SAFER maintains well-separated task geometry while producing a unified residual-based score that simultaneously supports task selection and unknown detection under the no-replay constraint.

The main contributions are outlined as follows:

**Perspective.** We reveal that the closed-world assumption in existing GCIL methods leads to catastrophic overconfidence due to unsupervised representation drift, and we propose a replay-free open-set GCIL setting to bridge this gap.

**Framework.** We propose SAFER, a unified framework with elegant theoretical guarantees. It utilizes drift-resilient subspace fingerprints and space-consistency regularization to reshape the features, achieving robust task routing and unknown rejection.

**Performance.** We demonstrate that SAFER establishes a new state-of-the-art with 0.0% catastrophic forgetting under a no-replay constraint, while boosting open-set detection (OSCR) by up to 26.9% compared to the strongest baselines.

## 2. Related Work

### 2.1. Continual graph learning

Continual Graph Learning (CGL). To alleviate catastrophic forgetting in continual graph learning, existing approaches (He et al., 2023; Liu et al., 2021; Perini et al., 2022; Sun et al., 2023; Wang et al., 2022; Zhang et al., 2023a; 2022b; 2023b; Zhou & Cao, 2021) can be broadly grouped into three categories: regularization-based methods, replay-based methods, and parameter-isolation methods. TWP (Liu et al., 2021) regularizes training by leveraging local graph topology to identify parameters critical to neighborhood aggregation and constraining their updates to preserve topological knowledge. PUMA (Liu et al., 2024) leverages graph condensation (Yan et al., 2026) to compress historical graphs into compact synthetic graphs, reducing memory overhead while mitigating catastrophic forgetting. Parameter isolation-based approaches, such as PI-GNN (Zhang et al., 2023a), allocate separate sets of parameters to different tasks to avoid inter-task interference. In addition, TPP (Niu et al., 2024b) reduces GCIL to a GTIL problem, achieving near-100% task-ID prediction accuracy and 0% forgetting. Despite their effectiveness in mitigating forgetting, these methods typically assume that all test-time samples belong to previously observed classes; when unknown classes appear, they can produce overconfident predictions.

### 2.2. Graph Open-Set Recognition

GOSR (Dong et al., 2025) aims to classify known classes while detecting samples from unseen classes, requiring the model to adapt beyond the closed training label space. The Entropy Message Passing (EMP) mechanism (Yang et al., 2023) quantifies unknownness by propagating entropy with structural cues and separating nodes via entropy-based clustering. EGonc (Zhang et al., 2024a) also studies open-set node classification but under different settings, which synthesizes substitute unknown samples and uses energy logits derived from feature residuals to a principal space. In contrast, we tackle a more challenging continual setting: under a no-replay constraint, the model must simultaneously mitigate forgetting and achieve reliable rejection and routing for unknown classes.

# 3. Methodology

## 3.1. Problem Formulation

**Graph Class-Incremental Learning (GCIL).** Considering a continuous learning scenario where a model is trained on a sequence of $T$ tasks, denoted as $\{\mathcal{T}^1, \mathcal{T}^2, \ldots, \mathcal{T}^T\}$. Each task $\mathcal{T}^t$ arrives with a task-specific graph $\mathcal{G}^t = (\mathcal{V}^t, \mathcal{E}^t)$, where $\mathcal{V}^t$ is the set of nodes, $\mathcal{E}^t$ represents the edges described by adjacency matrix $A^t$, $X^t \in \mathbb{R}^{|\mathcal{V}^t| \times d}$ denotes node features of $\mathcal{V}^t$, and $Y^t$ is the label set for task $t$. A key characteristic of GCIL is that the label sets are mutually disjoint, i.e., $Y^i \cap Y^j = \varnothing, \forall i \neq j$. During the training phase of task $t$, the model has access only to the current dataset $\mathcal{G}^t$, while access to full historical data $\{\mathcal{G}^1, \ldots, \mathcal{G}^{t-1}\}$ is strictly prohibited. The canonical goal of GCIL is to learn a mapping $f_\theta : \mathcal{V} \rightarrow \mathcal{Y}_{\leq t}$ that minimizes classification error on the cumulative label set $\mathcal{Y}_{\leq t} = \bigcup_{i=1}^{t} Y^i$.

**Open-Set evaluation protocol.** Standard GCIL assumes that test samples are strictly drawn from $\mathcal{Y}_{\leq t}$. In the Open-Set GCIL setting, we remove this assumption to align with the dynamic real-world environments. During inference at step $t$, the test graph consists of in-distribution nodes belonging to the observed classes $\mathcal{Y}_{\leq t}$ and out-of-distribution nodes from the unknown set. These unknown nodes may originate from future tasks or unrelated semantic spaces that have not yet been exposed.

## 3.2. Observation of Representation Drift

In this section, we use several simple and widely adopted softmax-thresholding baselines to diagnose open-set behavior in CGL. We find that, under future-task inputs, softmax confidence is unreliable because unknown samples can still be accepted with high confidence, and two diagnostic experiments further reveal the underlying failure chain.

From the outcome perspective, Fig.3 (left) shows that samples from a future task in two datasets still obtain high maximum softmax confidence over known classes, causing substantial overlap between the confidence distributions of known and unknown data; consequently, threshold-based rejection lacks a stable score margin. We then trace this behavior to its underlying cause. Fig.3 (right) demonstrates that, as the task stream progresses, the embedding centroid of an early task exhibits steadily increasing drift, indicating that continual updates keep rewriting the geometry of the representation space. Under such drift, unknown samples are more likely to be mapped into regions associated with high-confidence predictions for seen classes, leading to blindly overconfident misclassification.

The issue is not merely an ill-chosen threshold, but a fundamental limitation: standard CGL lacks a cross-task comparable and accumulative criterion for open-set decision making, even though open-set discrimination requires a stable reference under a drifting representation space. Motivated by this, **SAFER** introduces drift-resilient subspace fingerprints with structure-aware feature reshaping to unify task routing and open-set rejection, and further enforces geometric space-consistency regularization to improve intra-class compactness and suppress cross-task drift.

## 3.3. Drift-resilient Fingerprint Construction

Motivated by the drift-induced overconfidence diagnosed in Figure 3, SAFER seeks a cross-task comparable criterion that remains stable throughout the task stream. Therefore, we decouple the reference used for decision making from the plastic parameters used to learn new tasks. We maintain a pretrained GNN $f_\theta$ as the anchor network whose parameters are never updated when meeting new tasks. It provides a stable representational coordinate system throughout the entire task sequence, while leaving plasticity to lightweight adaptation modules introduced later (see Section 3.5). This prevents the reference for open-set decision making from being repeatedly rewritten by continual training.

**Structure-aware signal enhancement.** For a graph $\mathcal{G} = (\mathcal{V}, \mathcal{E})$ with node feature matrix $X$, node semantics are often coupled with local connectivity structure. Using only the raw features $X$ can cause representations to drift across tasks due to noise or heterophilous edges. To inject a stable structural prior into the input space prior to fingerprint construction, we apply a PPR-style (Personalized PageRank style) diffusion to $X$. Define the propagation operator $\tilde{A} = AD^{-1}$ and iterate:

$$H^{(k+1)} = (1 - \alpha)\tilde{A}H^{(k)} + \alpha X \qquad (1)$$

where $H^{(0)} = X$, $k = 0, \ldots, K-1$, $\alpha$ denotes the propagation weights. We then inject the diffused features into the original features via a residual form: $X^{\text{ppr}} = X + \eta H^{(K)}$. This module is applied consistently across all tasks, providing a cross-task consistent geometric basis. Therefore, the anchor network outputs node representations and applies $\ell_2$ normalization as:

$$z_v = \frac{f_\theta(\mathcal{G}, X^{\text{ppr}})_v}{\|f_\theta(\mathcal{G}, X^{\text{ppr}})_v\|_2} \in \mathbb{R}^d \qquad (2)$$

where normalization constrains representations to the unit sphere, making distance measurements across tasks more stable and avoiding spurious drift caused solely by scale changes.

**Geometric subspace fingerprints.** Upon the completion of task $t$, a geometric fingerprint is constructed to compress the task distribution into a compact, comparable manifold using the anchor embeddings. Let $\mathcal{V}_t$ denote the set of

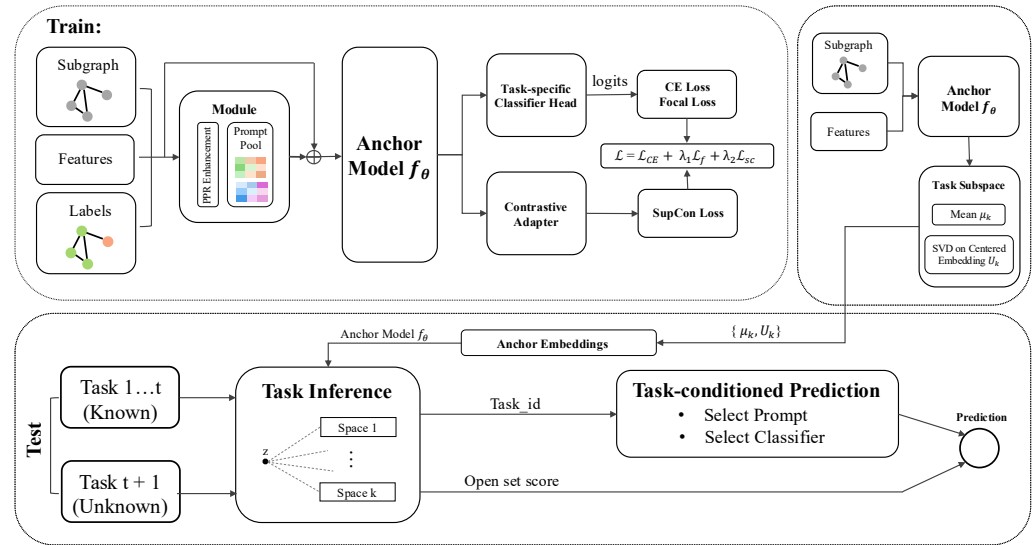

*Figure 2.* Overall framework. After each task, we store its center and low-rank subspace as a fingerprint in memory. We use the consistency energy to both route a query to the most compatible seen task and reject it as unknown when the score exceeds a threshold. If accepted, we apply the corresponding lightweight adaptation and task head to predict.

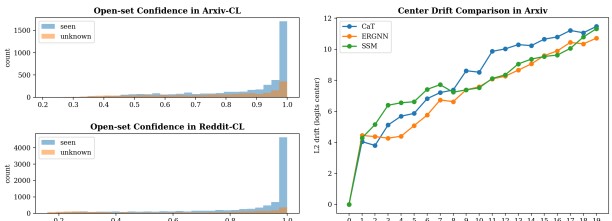

*Figure 3.* Left: train on several tasks and test on unknown tasks to show max-softmax confidence in Arxiv and Reddit. Right: Task-1 centroids drift with new tasks in the three classic GCIL methods.

training nodes for the current task. First, the distribution is anchored by computing the normalized task centroid $\mu_t$:

$$\mu_t = \frac{1}{|\mathcal{V}_t|} \sum_{v \in \mathcal{V}_t} z_v, \qquad \mu_t \leftarrow \frac{\mu_t}{\|\mu_t\|_2} \tag{3}$$

However, a centroid alone is insufficient to capture the complex within-task geometry. To characterize the task-specific geometric shape, we seek an optimal orthonormal basis $U_t \in \mathbb{R}^{d \times r}$ that maximizes the preserved signal energy of the node embeddings. Formally, this yields the following trace maximization problem,

$$\begin{aligned} U_t &= \underset{U \in \mathbb{R}^{d \times r}, U^\top U = I_r}{\arg\max} \sum_{v \in \mathcal{D}_t} \|U^\top (z_v - \mu_t)\|_2^2 \\ &= \underset{U \in \mathbb{R}^{d \times r}, U^\top U = I_r}{\arg\max} \mathrm{Tr}\left(U^\top \Sigma_t U\right), \end{aligned} \tag{4}$$

where $\Sigma_t = \mathbb{E}[(z - \mu_t)(z - \mu_t)^\top]$ is the task-specific covariance matrix. It is well-established that the optimal solution to this objective is given by the top-$r$ eigenvectors of the

covariance matrix $\Sigma_t$, which corresponds to performing Truncated PCA. The resulting task fingerprint is formally defined as $\mathcal{S}_t = (\mu_t, U_t)$. By encapsulating both location $\mu_t$ and structural orientation $U_t$, the fingerprints explicitly characterize the low-rank geometry of task $t$ to provide a static, drift-resilient reference for inference, remaining invariant as new tasks emerge.

Note that the storage overhead of the fingerprints is minimal, and they can be updated incrementally for each task. Geometrically, this is equivalent to approximating each task distribution as an affine subspace, representing the seen tasks as a collection of manifolds $\{\mathcal{S}_1, \ldots, \mathcal{S}_t\}$:

$$\mathcal{S}_t \triangleq \mu_t + \mathrm{span}(U_t). \tag{5}$$

### 3.4. Unified Inference via Feature Reshaping

Leveraging the drift-resilient fingerprints $\mathcal{S}_t$ constructed above, we introduce the feature reshaping mechanism for the inference to overcome the limitations of standard metric learning in open-set scenarios. Since distribution shift often distorts feature magnitudes in the original space, direct metric comparisons become unstable. Instead, our mechanism dynamically aligns test-time queries with each candidate task manifold via geometric projection, robustly evaluating the conformity of a query to task-specific constraints.

**Reshaping via projection.** Given a query embedding $q$ computed via Eq. (2) for a test node, the inference process involves translating the query toward each task centroid and reconstructing a task-conditioned representation. This

process treats the task fingerprint not merely as a reference point, but as a rigid geometric filter. For a candidate task $t$, let $\Delta_t = q - \mu_t$ and $\hat{\Delta}_t = U_t U_t^\top \Delta_t$. The reshaped embedding $\hat{q}_t$ is obtained by projecting the deviation onto the task manifold:

$$\hat{q}_t = \mu_t + \hat{\Delta}_t = \mu_t + U_t U_t^\top (q - \mu_t). \qquad (6)$$

This reconstruction explicitly discards the components of $q$ that are orthogonal to $\mathrm{span}(U_t)$. Geometrically, this operation filters out noise or drift-induced variations that are inconsistent with task $t$'s low-rank structure, retaining only the valid, task-relevant signal.

**Energy-based criterion for inference.** The magnitude of the discarded component during the reshaping can serve as a direct measure of the validity of the query with respect to the task geometry, enabling robust rejection of outliers that violate this subspace constraint. Based on this insight, we propose a unified energy score $D_t(q)$ to quantify the compatibility between the query and a task. This energy is designed to penalize the structural shape misalignment and positional location shift simultaneously:

$$D_t(q) = \|(I - U_t U_t^\top)(q - \mu_t)\|_2^2 + \lambda \|q - \mu_t\|_2^2. \quad (7)$$

The first term measures the reconstruction residual, and the second term is a regularization that prevents false positives from distant samples that might coincidentally lie on the subspace plane. Particularly, the former is strictly equivalent to the Euclidean distance from the query to the infinite affine subspace $\mathcal{S}_t$ as formalized in Proposition 3.1.

**Proposition 3.1.** *Let $\mathcal{S}_t = \mu_t + \mathrm{span}(U_t)$ with $U_t^\top U_t = I$. Then the projection residual is exactly the distance to the set $d(q, \mathcal{S}_t) \triangleq \inf_{u \in \mathcal{S}_t} \|q - u\|_2 = \left\|(I - U_t U_t^\top)(q - \mu_t)\right\|_2$.*

For inference, we identify the best-matching task by minimizing $D_t(q)$, while simultaneously using the energy value itself as an uncertainty measure. We define the optimal task index $\hat{\tau}(q)$ and the minimum energy score $s(q)$ as:

$$\hat{\tau}(q) = \arg\min_{i \leq T_{\text{seen}}} D_i(q), \quad s(q) = \min_{i \leq T_{\text{seen}}} D_i(q). \quad (8)$$

Based on these metrics, we formulate a ***hierarchical routing decision***. Given a decision threshold $\delta$, the routing outcome $\mathcal{R}(q)$ is defined as:

$$\mathcal{R}(q) = \begin{cases} \text{Unknown,} & \text{if } s(q) > \delta, \\ \text{Task } \hat{\tau}(q), & \text{otherwise.} \end{cases} \quad (9)$$

Here, $\hat{\tau}(q)$ performs ***task routing*** to select the target classifier, while $s(q)$ serves as the ***open-set uncertainty score***. Clearly, a lower energy indicates higher confidence that the query belongs to the known distribution. Under the class-incremental protocol, after completing task $t$, samples from

$\mathcal{Y}_{\leq t}$ are treated as known $\mathcal{K}$, and samples from task $t+1$ are treated as unknown $\mathcal{U}$; we evaluate unknown detection using $s(\cdot)$.

**Theoretical guarantees.** We provide several theoretical analyses to demonstrate that the proposed energy-based criterion $D_t(q)$ is robust for both rejection and routing.

Regarding open-set rejection, when the high-probability mass of known queries lies near its true task fingerprint, while that of unknown queries remains separated from all seen task fingerprints, the proposition 3.2 demonstrates that the energy score $\min D_i(\cdot)$ creates a provable margin between known and unknown distributions.

**Proposition 3.2** (Separation). *Let $q \sim P_{\mathcal{K}}$ be a known query with true task $y$, and let $q' \sim P_{\mathcal{U}}$ be an unknown query. Recall that the task-wise energy is defined as Eq. (7) and the unified inference score is $s(x) = \min_{1 \leq i \leq T_{\text{seen}}} D_i(x)$. For an unknown query $q'$, define $m_{\mathcal{S}}(q') := \min_{1 \leq i \leq T_{\text{seen}}} d(q', \mathcal{S}_i)$ and $m_\mu(q') := \min_{1 \leq i \leq T_{\text{seen}}} \|q' - \mu_i\|_2$. Assume there exist constants $R_{\mathcal{S}}, R_\mu, \Delta_{\mathcal{S}}, \Delta_\mu > 0$ and tolerances $\alpha, \beta \in (0, 1)$, such that:*

$$\Pr_{q \sim P_{\mathcal{K}}} \left( d(q, \mathcal{S}_y) \leq R_{\mathcal{S}}, \ \|q - \mu_y\|_2 \leq R_\mu \right) \geq 1 - \alpha \quad (10)$$

$$\Pr_{q' \sim P_{\mathcal{U}}} \left( m_{\mathcal{S}}(q') \geq \Delta_{\mathcal{S}}, \ m_\mu(q') \geq \Delta_\mu \right) \geq 1 - \beta \quad (11)$$

*If the probabilistic margin condition $\Delta_{\mathcal{S}}^2 + \lambda \Delta_\mu^2 > R_{\mathcal{S}}^2 + \lambda R_\mu^2$ holds, then for any threshold $\delta$ satisfying*

$$R_{\mathcal{S}}^2 + \lambda R_\mu^2 < \delta < \Delta_{\mathcal{S}}^2 + \lambda \Delta_\mu^2 \quad (12)$$

*the unified score $s(\cdot)$ satisfies $\Pr_{q \sim P_{\mathcal{K}}}(s(q) > \delta) \leq \alpha$ and $\Pr_{q' \sim P_{\mathcal{U}}}(s(q') \leq \delta) \leq \beta$. Consequently, under a single threshold $\delta$, the false rejection rate on known queries is at most $\alpha$, and the false acceptance rate on unknown queries is at most $\beta$.*

Beyond rejection, we also require the correct task routing. For a known query $q$, we want $\arg\min_i D_i(q)$ to select the true task $y$. The Proposition 3.3 provides an explicit lower bound on the energy gap, showing that routing margin is jointly controlled by between-task separability and within-task compactness.

**Proposition 3.3** (Routing Margin). *Let $q$ be a query belonging to the true task $y$, satisfying the intra-task compactness bounds $d(q, \mathcal{S}_y) \leq R_{\mathcal{S}}$ and $\|q - \mu_y\|_2 \leq R_\mu$. Define the minimum inter-task separability as $D_{\mathcal{S}} \triangleq \min_{c \neq y} d(\mathcal{S}_y, \mathcal{S}_c)$ and $D_\mu \triangleq \min_{c \neq y} \|\mu_y - \mu_c\|_2$. Then, for any incorrect task $c \neq y$, the energy gap is lower-bounded by:*

$$\begin{aligned} D_c(q) - D_y(q) \geq\ & \left[(D_{\mathcal{S}} - R_{\mathcal{S}})^2 - R_{\mathcal{S}}^2\right] \\ & + \lambda \left[(D_\mu - R_\mu)^2 - R_\mu^2\right]. \end{aligned} \quad (13)$$

Proposition 3.3 reveals that the robustness of task routing relies on the trade-off between inter-task separability $(D_\mathcal{S}, D_\mu)$ and intra-task compactness $(R_\mathcal{S}, R_\mu)$. Hence, when between-task separation $(D_\mathcal{S}, D_\mu^{\text{near}})$ is sufficiently large and within-task compactness $(R_\mathcal{S}, R_\mu)$ is sufficiently small, the $\arg\min$ routing induced by the energy has a larger margin and is therefore more robust.

### 3.5. Optimization with Geometric Consistency

Guided by Proposition 3.3, robust routing requires minimizing intra-task radii $R_\mathcal{S}$ and $R_\mu$, yet directly updating the backbone distorts historical manifolds and triggers the catastrophic drift observed. To resolve this, we propose a paradigm shift to Input-Level Adaptation. Rather than altering parameters to fit evolving data, we treat the frozen anchor as a universal coordinate system. By dynamically modulating input representations via prompts and adapters, we reshape diverse task distributions to align with this stable geometry, thereby accommodating new tasks without overwriting the global routing map.

**Node-adaptive prompts.** To adapt to incoming tasks $(t > 1)$ without altering the frozen anchor parameters $\theta$, we intervene at the input level using learnable prompts. Unlike static prompting strategies, a node-adaptive prompting mechanism is employed to handle the structural diversity of graph data.

Specifically, we maintain a task-specific prompt bank $P_t \in \mathbb{R}^{k \times d}$, consisting of $k$ learnable prompt tokens (or basis vectors). Given the enhanced features $X^{\text{ppr}}$, the model dynamically constructs a unique prompt for each node by querying this bank. An attention mechanism first computes the relevance scores between the node content and the prompt tokens:

$$\text{Attn}(X^{\text{ppr}}) = \text{softmax}(X^{\text{ppr}}W) \in \mathbb{R}^{N \times k}, \quad (14)$$

where $W \in \mathbb{R}^{d \times k}$ is a lightweight projection matrix. The final input representation is then synthesized by injecting a weighted combination of the prompt tokens:

$$\tilde{X} = X^{\text{ppr}} + \text{Attn}(X^{\text{ppr}})P_t. \quad (15)$$

This formulation can be interpreted as a conditional feature transformation: rather than applying a rigid shift, each node essentially selects the most optimal combination of prompts to align itself with the frozen anchor space. This design confines task-specific plasticity to a minimal set of parameters $(P_t, W)$, enabling effective adaptation to new distributions while preserving the stability of the geometric fingerprints.

**Adapter-mediated geometric regularization.** Cross-entropy loss focuses solely on decision boundaries and does not guarantee the tight intra-class clustering required to minimize the radii $R_\mathcal{S}$ and $R_\mu$. To explicitly enforce the compactness condition derived in Proposition 3.3, we introduce

a geometric regularization term. Crucially, applying such constraints directly to the backbone representations $z_v$ can disrupt the pre-trained semantic manifold. To decouple the metric learning objective from the general semantic space, we introduce a lightweight adapter $\phi_t$ that maps representations into a task-specific metric hypersphere $\tilde{z}_v = \frac{\phi_t(z_v)}{\|\phi_t(z_v)\|_2}$. We then apply the supervised contrastive loss (Khosla et al., 2020) explicitly on these projected embeddings:

$$\mathcal{L}_{\text{supcon}} = -\sum_{i \in \mathcal{B}} \frac{1}{|\mathcal{P}(i)|} \sum_{p \in \mathcal{P}(i)} \log \frac{\exp(\tilde{z}_i^\top \tilde{z}_p / \tau)}{\sum_{a \in \mathcal{A}(i)} \exp(\tilde{z}_i^\top \tilde{z}_a / \tau)}. \quad (16)$$

where $\mathcal{B}$ denotes the mini-batch indices, $\mathcal{P}(i) = \{p \in \mathcal{B} \setminus \{i\} \mid y_p = y_i\}$ is the set of positive samples, $\mathcal{A}(i) = \mathcal{B} \setminus \{i\}$ is the set of all other samples, and $\tau > 0$ is a scalar temperature parameter.

In this decoupled architecture, the adapter $\phi_t$ identifies geometric discrepancies in the metric space and guides the prompt $P_t$ to modulate input features such that they naturally form compact clusters, minimizing $R_\mathcal{S}$ and $R_\mu$ without distorting the backbone's intrinsic knowledge.

To perform within-task classification, we also maintain a linear classifier head $g_t : \mathbb{R}^{d_h} \to \mathbb{R}^C$ and the prediction is given by $\hat{y} = g_t(z_v)$, where $C$ is the number of classes. We use class-weighted cross-entropy $\mathcal{L}_{\text{ce}}$, and optionally employ a focal term to emphasize hard examples and suppress gradients dominated by easy examples. Therefore, the final training objective for task $t$ balances classification accuracy and geometric compactness:

$$\mathcal{L}_{\text{total}} = \mathcal{L}_{\text{ce}} + \lambda_{\text{im}}\mathcal{L}_{\text{focal}} + \lambda_{\text{geo}}\mathcal{L}_{\text{supcon}}, \quad (17)$$

where $\lambda_{\text{im}}$ and $\lambda_{\text{geo}}$ are hyperparameters. This composite objective ensures that the model not only learns accurate decision boundaries but also explicitly shapes the feature space to maximize the inference margin.

## 4. Experiments

**Datasets.** Following the GCLB (Zhang et al., 2023a), four public graph datasets are employed, including CoraFull (Mc-Callum et al., 2000), Arxiv (Hu et al., 2020), Reddit (Hamilton et al., 2017) and Products (Hu et al., 2020). Specifically, CoraFull and Arxiv are citation networks, Reddit is derived from posts on the Reddit platform, and Products is a co-purchasing network extracted from Amazon. For all datasets, each task includes two classes. Moreover, for each class, the data are split into training, validation, and testing sets with proportions of 0.6, 0.2 and 0.2 respectively.

**Baselines.** We compare our model against two categories of SOTA continual learning baselines: (1) General CIL methods: EWC (Kirkpatrick et al., 2017), LwF (Li & Hoiem, 2017), GEM (Lopez-Paz & Ranzato, 2017) and

*Table 1.* **Results on CoraFull, Arxiv, Reddit, and Products.** Baseline methods under the GCIL setting are grouped according to their methodological categories. "↑" indicates that higher is better. Oracle has access to data from all tasks and the ground-truth task IDs at inference, thus providing an upper bound. "NA" denotes not applicable. The shaded row highlights our method.

| Category | Methods | CoraFull AP↑ | AF↑ | AUC↑ | OSCR↑ | Arxiv AP↑ | AF↑ | AUC↑ | OSCR↑ | Reddit AP↑ | AF↑ | AUC↑ | OSCR↑ | Products AP↑ | AF↑ | AUC↑ | OSCR↑ |
|---|---|---|---|---|---|---|---|---|---|---|---|---|---|---|---|---|---|
| Lower bound | Fine-tune | 3.5 | -95.2 | 53.9 | 7.4 | 4.9 | -89.7 | 50.5 | 12.4 | 5.9 | -97.9 | 55.0 | 6.7 | 7.6 | -88.7 | 60.4 | 4.5 |
| Upper bound | Oracle | 95.5 | NA | NA | NA | 90.3 | NA | NA | NA | 99.5 | NA | NA | NA | 95.3 | NA | NA | NA |
| Regularization | EWC [PNAS'17] | 52.6 | -38.5 | 46.5 | 3.0 | 8.5 | -69.6 | 49.6 | 0.0 | 10.3 | -33.2 | 65.3 | 16.0 | 23.8 | -21.7 | 43.7 | 11.6 |
| | MAS [ECCV'18] | 6.5 | -92.3 | 56.4 | 0.0 | 4.8 | -72.2 | 49.9 | 0.0 | 9.2 | -23.1 | 65.0 | 16.7 | 16.7 | -57.0 | 42.2 | 10.7 |
| | GEM [NeurIPS'17] | 8.4 | -88.4 | 53.8 | 9.9 | 4.9 | -89.8 | 48.8 | 0.0 | 11.5 | -92.4 | 64.1 | 18.1 | 4.5 | -94.7 | 49.4 | 10.7 |
| | TWP [AAAI'18] | 62.6 | -30.6 | 63.7 | 21.7 | 6.7 | -50.6 | 57.8 | 19.4 | 8.0 | -18.8 | 54.9 | 2.4 | 14.1 | -11.4 | 52.3 | 22.9 |
| Distillation | LwF [TPAMI'18] | 33.4 | -59.6 | 51.9 | 10.1 | 9.9 | -43.6 | 52.5 | 17.5 | 86.6 | -9.2 | 61.1 | 15.3 | 48.2 | -18.6 | 47.2 | 10.6 |
| Replay | ERGNN [AAAI'21] | 34.5 | -61.6 | 48.4 | 6.9 | 21.5 | -70.0 | 51.3 | 19.1 | 82.7 | -17.3 | 71.5 | 64.3 | 48.3 | -45.7 | 40.2 | 17.0 |
| | SSM-uniform [ICDM'22] | 73.0 | -14.8 | 35.8 | 63.3 | 47.1 | -11.7 | 46.5 | 35.0 | 94.3 | -1.4 | 74.8 | 76.2 | 62.0 | -9.9 | 50.3 | 50.8 |
| | SSM-degree [ICDM'22] | 75.4 | -9.7 | 50.0 | 3.5 | 48.3 | -10.7 | 46.3 | 35.3 | 94.4 | -1.3 | 75.0 | 76.3 | 63.3 | -9.6 | 75.0 | 76.3 |
| | SEM-curvature [TNNLS'23] | 77.7 | -10.0 | - | - | 49.9 | -8.4 | - | - | 96.3 | -0.6 | - | - | 65.1 | -9.5 | - | - |
| | CaT [ICDM'23] | 80.4 | -5.3 | 70.4 | 63.0 | 48.2 | -12.6 | 66.9 | 59.3 | 97.3 | -0.1 | 72.6 | 81.3 | 70.3 | -4.5 | 54.4 | 61.5 |
| | DeLoMe [ECAI'24] | 81.0 | -3.3 | 84.9 | 80.0 | 50.6 | 5.1 | 58.8 | 43.1 | 97.4 | 0.1 | 92.9 | 92.3 | 67.5 | -17.3 | 75.5 | 58.9 |
| | DMSG [ICLR'25] | 77.8 | -0.5 | 65.1 | 60.9 | 50.7 | -1.9 | 54.0 | 37.8 | 98.1 | 0.9 | 56.0 | 55.1 | 66.0 | -0.9 | 62.2 | 38.6 |
| | PUMA [TKDE'25] | 73.6 | -7.0 | 68.4 | 50.6 | 69.3 | -10.7 | 61.1 | 37.4 | 98.0 | -0.3 | 68.9 | 48.0 | 74.2 | -4.1 | 53.7 | 35.9 |
| Task-aware | ETPP [NeurIPS'24] | 93.4 | 0.0 | 79.8 | 78.6 | 85.4 | 0.0 | 59.3 | 58.1 | 99.5 | 0.0 | 71.0 | 70.9 | 94.0 | 0.0 | 69.4 | 68.8 |
| | DTPP [NeurIPS'24] | 93.4 | 0.0 | 59.7 | 57.8 | 85.4 | 0.0 | 68.7 | 65.5 | 99.5 | 0.0 | 69.6 | 69.5 | 94.0 | 0.0 | 69.4 | 67.8 |
| | MTPP [NeurIPS'24] | 93.4 | 0.0 | 75.3 | 73.7 | 85.4 | 0.0 | 68.4 | 65.9 | 99.5 | 0.0 | 72.5 | 72.5 | 94.0 | 0.0 | 78.5 | 77.4 |
| Subspace-aware | **SAFER (Ours)** | **94.9** | **0.0** | **97.9** | **94.6** | **90.6** | **0.0** | **100.0** | **92.4** | **99.5** | **0.0** | **100.0** | **99.6** | **95.0** | **0.0** | **100.0** | **97.1** |

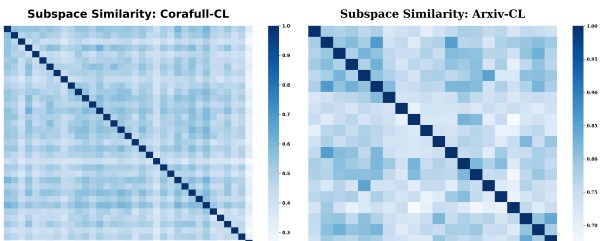

*Figure 4.* Subspace similarity matrices on CoraFull and Arxiv.

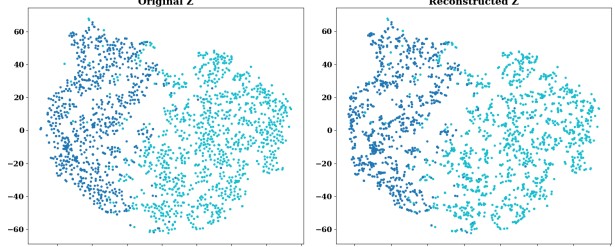

*Figure 5.* visualization of original vs. subspace-reconstructed anchor embeddings.

MAS (Aljundi et al., 2018) ; (2) Graph CIL methods: ERGNN (Zhou & Cao, 2021), TWP (Liu et al., 2021), SSM (Zhang et al., 2022b), SEM (Zhang et al., 2023b), CaT (Liu et al., 2023), DMSG (Qiao et al., 2024), DeLoMe (Niu et al., 2024a), PUMA(Liu et al., 2024) and TPP (Niu et al., 2024b). In addition, we include two further methods: Fine-Tune and Oracle. The Fine-Tune baseline incrementally fine-tunes the model from previous tasks without employing any continual learning strategies. The Oracle baseline (Niu et al., 2024b) assumes access to all task graphs jointly and performs GCL on the aggregated dataset; at inference time, it also uses the ground-truth task ID for each test sample.

### 4.1. Overall Performance Comparison

Compared with existing methods, Table 1 reports the main GCIL results on four datasets, including closed-set metrics Average Performance (AP) and Average Forgetting (AF), as well as open-set metrics area under the ROC curve (AUROC, AUC) and open-set classification rate (OSCR)(Dhamija et al., 2018). Since most GCIL baselines are not designed for open-set evaluation, we adopt a unified protocol and ad-

ditionally compute a consistent energy-style unknownness score for all methods, making AUROC/OSCR comparable. For TPP, we further report multiple scoring variants Energy-TPP (ETPP), Distance-TPP (DTPP) and Mix-TPP (MTPP). Overall, our method achieves the best performance across all datasets, with especially strong gains on open-set metrics.

1) Fine-tuning only on the current task causes clear degradation, showing that new-task gradients quickly overwrite earlier knowledge. 2) Baselines from Euclidean CIL exhibit large variance and often poor AP/AF, confirming that GCIL requires graph-aware solutions. Replay-based methods with a memory buffer are generally stronger in AP and forgetting control, suggesting that rehearsal is effective in retaining past information. However, even with our unified open-set scoring, these methods still tend to produce overconfident predictions on future-task samples. 3) TPP can achieve near-zero AF, validating that prompt-based parameter isolation is effective for forgetting mitigation. Under our unified protocol, however, no open-set scoring variant consistently

*Table 2.* Ablation results on CoraFull and Products.

| Setting | CoraFull | | | | Products | | | |
|---|---|---|---|---|---|---|---|---|
| | AP | AF | AUC | OSCR | AP | AF | AUC | OSCR |
| Full model | **94.9** | **0.0** | **97.9** | **94.6** | **95.0** | **0.0** | **100.0** | **97.1** |
| w/o Routing | 2.0 | -0.3 | 51.7 | 4.4 | 2.8 | -1.2 | 45.5 | 6.8 |
| Single classifier head | 89.3 | -3.4 | 97.64 | 90.49 | 80.9 | -13.6 | 100.0 | 78.5 |
| w/o Prompt, w/o PPR | 88.9 | 0.0 | 97.9 | 89.5 | 93.2 | 0.0 | 100.0 | 95.8 |
| w/o PPR | 94.8 | 0.0 | 96.2 | 93.3 | 93.5 | 0.0 | 100.0 | 95.8 |
| w/o Prompt | 89.2 | 0.0 | 96.2 | 88.1 | 93.5 | 0.0 | 100.0 | 95.8 |

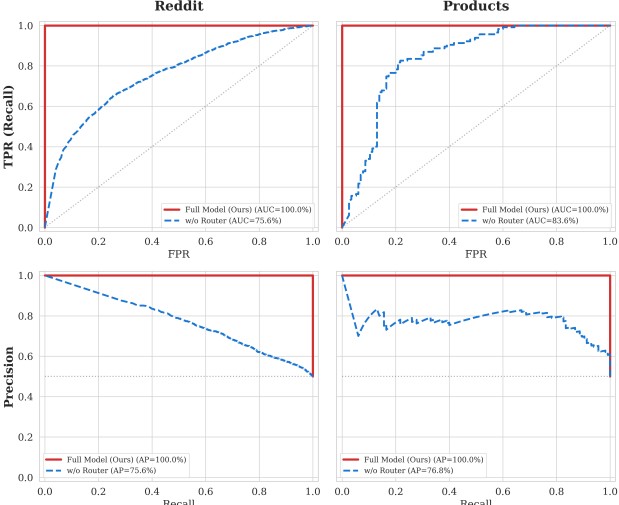

*Figure 6.* ROC/PR curves for unknown detection(full model vs. w/o Router) on Reddit and Products.

matches our performance across datasets. This suggests that accurate task-ID inference or parameter isolation alone is insufficient for reliable unknown rejection.

4) We achieve zero forgetting while attaining the best AP on two datasets. More importantly, we obtain consistently higher AUROC/OSCR, supporting that fingerprint-based consistency energy provides a stable and cross-task comparable signal: it enables robust routing when task IDs are unavailable and yields a reliable score for rejecting future-task/novel-class samples. 5) Oracle assumes access to all previously seen data and provides the task ID at inference, and thus represents an approximate upper bound for closed-set classification. Our approach achieves AP close to Oracle on multiple datasets, while maintaining clear open-set gains.

### 4.2. Ablation Studies

**Impact of Routing and Input Reshaping.** From Table 2, routing and predicting the task ID are critical under the GCIL setting. Removing it causes a pronounced drop in both closed-set and open-set performance, indicating that classifier outputs alone cannot reliably separate known from unknown samples. With routing preserved, Prompt and the task-specific head further increase cross-task subspace separability, making different $U_t$ more independent. Meanwhile, PPR promotes intra-class contraction and suppresses within-

*Table 3.* Comparison of reconstruction residual gaps on known and unknown samples.

| Dataset | Known $R_\mu$ | Unknown $R_\mu$ | Gap $R_\mu$ | Gap $R_S$ |
|---|---|---|---|---|
| CoraFull | 0.156 | 0.505 | 0.343 | 0.155 |
| Arxiv | 0.043 | 0.478 | 0.426 | 0.079 |
| Reddit | 0.011 | 0.664 | 0.651 | 0.244 |
| Products | 0.117 | 0.449 | 0.803 | 0.422 |

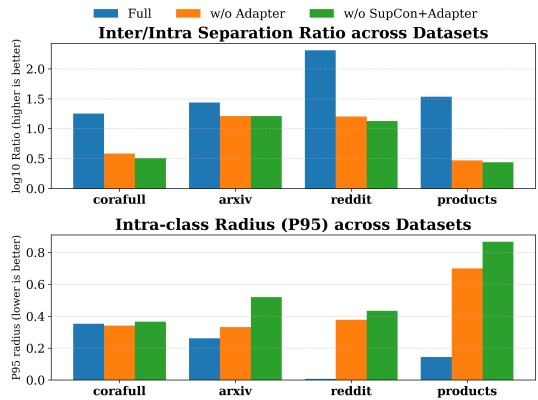

*Figure 7.* Cluster geometry on four datasets: $\log_{10}(d_{\text{inter}}/d_{\text{intra}})$ and intra-class P95 radius.

class noise, yielding more concentrated residual energies and thus more stable unknown rejection.

**Effect of Subspace Fingerprints.** The low-rank subspace $U_t$ constructs a long-term fingerprint for each task, and is used at inference time for task routing and unknown-score estimation. As shown in Fig. 4, the cross-task similarity $\text{Sim}(i,j) = \|U_i^\top U_j\|_F^2$ exhibits a "high-diagonal, low-off-diagonal" pattern, indicating that subspace directions of different tasks are well separated. Therefore, known samples are more likely to obtain smaller residual energies and be correctly matched, while unknown samples yield larger residuals, enabling reliable rejection.

**Feature Reshaping for Task-Aligned Representations.** As shown in Fig. 5, we further compare the original anchor representations with the reconstructed representations obtained after projection onto the task subspace. Compared with the original representations, the reconstructed representations exhibit a more pronounced intra-class clustering trend and clearer task boundaries in the spatial distribution, indicating that the feature reshaping mechanism can effectively filter out components irrelevant to the current task, aligning the query representations better with the candidate task manifold. Guided by task fingerprints for geometric alignment of test samples and combined with the reconstruction residuals for task routing and open-set discrimination, the model can stabilize the representation space, suppress cross-task drift, and improve the reliability of subsequent classification and rejection.

*Table 4.* **Results under different class-per-task settings.** Comparison of DMSG, TPP, and SAFER on four graph benchmarks.

| Dataset | Setting | Method | AP↑ | AF↑ | AUC↑ | OSCR↑ |
|---|---|---|---|---|---|---|
| CoraFull | 3 cls/task | DMSG | 76.7 | 2.4 | 65.6 | 61.0 |
| | | TPP | 90.4 | 0.0 | 81.4 | 77.7 |
| | | **SAFER** | **94.1** | **0.0** | **99.2** | **93.6** |
| | 5 cls/task | DMSG | 75.8 | 3.4 | 65.6 | 60.4 |
| | | TPP | 86.4 | 0.0 | 76.1 | 71.4 |
| | | **SAFER** | **90.9** | **0.0** | **99.5** | **91.2** |
| Arxiv | 3 cls/task | DMSG | 48.7 | 2.7 | 54.8 | 37.7 |
| | | TPP | 78.0 | 0.0 | 59.0 | 56.3 |
| | | **SAFER** | **88.8** | **0.0** | **100.0** | **87.6** |
| | 5 cls/task | DMSG | 45.5 | 4.2 | 53.7 | 35.8 |
| | | TPP | 62.8 | 0.0 | 65.0 | 58.5 |
| | | **SAFER** | **82.3** | **0.0** | **100.0** | **81.8** |
| Reddit | 3 cls/task | DMSG | 93.5 | -3.5 | 49.2 | 47.4 |
| | | TPP | 98.7 | 0.0 | 70.5 | 70.3 |
| | | **SAFER** | **99.2** | **0.0** | **100.0** | **99.5** |
| | 5 cls/task | DMSG | 91.8 | -5.0 | 42.4 | 39.7 |
| | | TPP | 97.7 | 0.0 | 71.4 | 71.0 |
| | | **SAFER** | **98.6** | **0.0** | **100.0** | **99.0** |
| Products | 3 cls/task | DMSG | 51.9 | -8.6 | 63.8 | 34.6 |
| | | TPP | 80.3 | 0.0 | 65.3 | 63.7 |
| | | **SAFER** | **91.7** | **0.0** | **100.0** | **95.7** |
| | 5 cls/task | DMSG | 49.5 | 3.5 | 64.5 | 35.7 |
| | | TPP | 70.3 | 0.0 | 61.2 | 58.2 |
| | | **SAFER** | **79.7** | **0.0** | **100.0** | **93.1** |

**Unknown detection via routing scores.** Fig.6 compares the ROC and PR curves for unknown detection on two datasets between the full model and the variant (w/o Router). The full model uses the subspace-consistency score and achieves nearly ideal detection performance on all datasets. In contrast, after removing the open-set router, the model falls back to the maximum softmax confidence of the classifier as the unknown-detection score, leading to a clear degradation of the curves; in particular, on Reddit, the AUROC drops from 100% to 75.6%. These results indicate that routing scores based on subspace fingerprints significantly improve the overall ranking quality for rejecting unknown samples while accepting known ones.

**Effect of Adapter and SupCon.** To evaluate the effect of the projection-based Adapter and SupCon on task-aligned representations, we measure cluster structure using two metrics aligned with Proposition 3.2: (i) the inter-/intra-class separation ratio $\text{Ratio} = d_{\text{inter}}/d_{\text{intra}}$, and (ii) the intra-class radius P95, i.e. the 95th-percentile intra-class distance. As shown in Fig. 7, introducing Adapter and SupCon enhances the learned representation space by increasing the separation ratio and reducing the P95 radius. Specifically, known-task samples become more tightly concentrated within their respective task subspaces, while unknown-task samples maintain a clear geometric gap from all seen-task subspaces. This effect compresses intra-task distances, tightens class clusters, and preserves inter-task separation, resulting in more stable task routing and robust open-set discrimination.

**Empirical Geometric Gap.** As shown in Table 3, we additionally measured the empirical geometric gap. We observe that the average minimum and of unknown samples with respect to seen task fingerprints are consistently and significantly larger than those of known samples. Moreover, the higher inter/intra separation ratio and the smaller P95 radius of the full model in Fig. 7 further indicate that SAFER indeed pushes the learned representation space toward the geometric regime required by Proposition 3.2.

### 4.3. Analysis under Additional Task Settings

**Robustness under different class-per-task settings.** Table 4 compares DMSG, TPP, and SAFER under different class-per-task settings on four datasets. As the number of classes per task increases from 2 to 3, and then to 5, all methods generally exhibit a decline in performance across AP, AF, AUC, and OSCR, reflecting the increased difficulty of incremental learning when each task contains more classes. Despite this challenge, SAFER consistently achieves the strongest performance on nearly all metrics, particularly on AUC and OSCR, demonstrating superior open-set discrimination. Compared with TPP, SAFER not only maintains stable incremental classification but also significantly improves unknown-class detection, with particularly large AUC and OSCR gains on Arxiv and Products under both settings. Overall, these observations suggest that SAFER's subspace-aware routing and feature reshaping mechanisms effectively preserve task-specific structure and mitigate cross-task interference, yielding robust and reliable performance even as task granularity increases.

## 5. Conclusion

In this paper, we identified that standard GCIL methods falter in open-world scenarios due to catastrophic representation drift and generalized overconfidence. To overcome these challenges, we proposed SAFER, a novel framework that shifts the paradigm from model-centric fine-tuning to subspace-aware input reshaping. By constructing drift-resilient geometric fingerprints, SAFER unifies task routing and open-set rejection into a single energy-based metric, achieving structural zero-forgetting. Extensive experiments demonstrate that SAFER establishes a new state-of-the-art in both closed-set accuracy and open-set detection.

Currently, SAFER operates under a fully supervised paradigm, relying on task-specific labels to construct precise subspace fingerprints. This dependence may limit its immediate applicability in label-scarce environments. In future work, we aim to extend this geometric framework to semi-supervised or self-supervised settings, exploring how to maintain robust open-set capabilities when label guidance is sparse or absent.

## Acknowledgements

This work was supported in part by the Scientific and Technological Innovation Project of China Academy of Chinese Medical Sciences ( CI2023C015YL), the National Natural Science Foundation of China (62506029, 62476022), Beijing Natural Science Foundation (4254085, L251043), and China Postdoctoral Science Foundation (2025M771589).

## Impact Statement

This paper presents work whose goal is to advance the field of Machine Learning. There are many potential societal consequences of our work, none of which we feel must be specifically highlighted here.

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

*Table 5.* Notations used in this paper.

| Notation | Definition |
|---|---|
| $\alpha$ | teleport coefficient in the diffusion update |
| $D_{\mathcal{S}}$ | Minimum subspace separability |
| $D_{\mu}$ | Minimum centroid separability |
| $D_t(q)$ | Energy of query $q$ w.r.t. task $t$ (Eq. (7)) |
| $d$ | Representation dimension |
| $d(q, \mathcal{S}_t)$ | Distance to task set |
| $K$ | Number of PPR diffusion iterations |
| $k$ | Number of prompt tokens. |
| $N$ | Number of nodes in the current graph |
| $s(q)$ | Open-set confidence score |
| $t$ | Task index |
| $T_{\text{seen}}$ | Number of seen tasks |
| $\hat{\tau}(q)$ | Task routing prediction |
| $\mathcal{E}$ | Edge set, $|\mathcal{E}| = m$ |
| $\mathcal{G} = (\mathcal{V}, \mathcal{E})$ | Graph with node set $\mathcal{V}$ and edge set $\mathcal{E}$ |
| $\mathcal{K}$ | Known sample set. |
| $\mathcal{S}_t$ | Task Subspace fingerprint |
| $\mathcal{U}$ | Unknown sample set |
| $\mathcal{V}$ | Node set. |
| $\mathcal{Y}_{\leq t}$ | Set of seen classes up to task $t$ |
| $d(\mathcal{S}_y, \mathcal{S}_c)$ | Distance between two task sets |
| $\Delta_t \in \mathbb{R}^d$ | Deviation from centroid |
| $\hat{\Delta}_t \in \mathbb{R}^d$ | Projected deviation: $\hat{\Delta}_t = U_t U_t^\top \Delta_t$ |
| $\hat{q}_t \in \mathbb{R}^d$ | Reshaped embedding for task $t$ |
| $\mu_t \in \mathbb{R}^d$ | Normalized task centroid |
| $q \in \mathbb{R}^d$ | Test-time query embedding |
| $\tilde{z}_v \in \mathbb{R}^d$ | The L2-normalized representation of task-specific adapter |
| $z_v \in \mathbb{R}^d$ | $\ell_2$-normalized node embedding |
| $A \in \mathbb{R}^{n \times n}$ | Adjacency matrix |
| $D \in \mathbb{R}^{n \times n}$ | Degree matrix |
| $f_\theta(\mathcal{G}, X^{\text{ppr}}) \in \mathbb{R}^{n \times d}$ | Anchor network output representation matrix |
| $H^{(k)} \in \mathbb{R}^{n \times d}$ | Diffused features at diffusion step $k$ |
| $U_t \in \mathbb{R}^{d \times r}$ | Orthonormal subspace basis for task $t$ |
| $W \in \mathbb{R}^{d \times k}$ | Projection matrix for prompt relevance |
| $X \in \mathbb{R}^{n \times d}$ | Raw node feature matrix |
| $X^{\text{ppr}} \in \mathbb{R}^{N \times d}$ | PPR-diffused node feature matrix |
| $\tilde{A} = AD^{-1} \in \mathbb{R}^{n \times n}$ | Propagation / normalization operator |
| $\tilde{X} \in \mathbb{R}^{N \times d}$ | Prompt-injected input features |

## A. Notation Table.

As an expansion of the notations in our work, we summarize the frequently used notations in Table 5.

## B. Proof of Propositions.

**Proposition B.1.** *Let $\mathcal{S}_t = \mu_t + \text{span}(U_t)$ with $U_t^\top U_t = I$. Then the projection residual is exactly the distance to the set* $d(q, \mathcal{S}_t) \triangleq \inf_{u \in \mathcal{S}_t} \|q - u\|_2 = \left\| (I - U_t U_t^\top)(q - \mu_t) \right\|_2$.

*Proof.* By definition, $d(q, \mathcal{S}_t) \triangleq \inf_{u \in \mathcal{S}_t} \|q - u\|_2$. Since $\mathcal{S}_t = \mu_t + \text{span}(U_t)$, any $u \in \mathcal{S}_t$ can be written as $u = \mu_t + U_t a$ for some $a \in \mathbb{R}^r$. Let $x \triangleq q - \mu_t$. Then $d(q, \mathcal{S}_t) = \inf_{a \in \mathbb{R}^r} \|x - U_t a\|_2$.

Consider $f(a) \triangleq \|x - U_t a\|_2^2$. Expanding and using $U_t^\top U_t = I_r$,

$$\begin{aligned}
f(a) &= (x - U_t a)^\top (x - U_t a) \\
&= x^\top x - x^\top U_t a - a^\top U_t^\top x + a^\top U_t^\top U_t a \\
&= \|x\|_2^2 - 2a^\top U_t^\top x + a^\top (U_t^\top U_t) a \\
&= \|x\|_2^2 - 2a^\top U_t^\top x + \|a\|_2^2.
\end{aligned}$$

Setting $\nabla_a f(a) = 0$ yields $a^\star = U_t^\top x$. Hence the closest point is $u^\star = \mu_t + U_t a^\star = \mu_t + U_t U_t^\top (q - \mu_t)$, and the residual is $r^\star = x - U_t a^\star = (I - U_t U_t^\top)(q - \mu_t)$. Therefore, $d(q, \mathcal{S}_t) = \|r^\star\|_2 = \left\| (I - U_t U_t^\top)(q - \mu_t) \right\|_2$, as claimed. $\qquad\square$

**Proposition B.2** (Separation). *Let $q \sim P_\mathcal{K}$ be a known query with true task $y$, and let $q' \sim P_\mathcal{U}$ be an unknown query. Recall that the task-wise energy is defined as Eq. (7) and the unified inference score is $s(x) = \min_{1 \leq i \leq T_{\text{seen}}} D_i(x)$. For an unknown query $q'$, define $m_\mathcal{S}(q') := \min_{1 \leq i \leq T_{\text{seen}}} d(q', \mathcal{S}_i)$ and $m_\mu(q') := \min_{1 \leq i \leq T_{\text{seen}}} \|q' - \mu_i\|_2$. Assume there exist constants $R_\mathcal{S}$, $R_\mu$, $\Delta_\mathcal{S}$, $\Delta_\mu > 0$ and tolerances $\alpha, \beta \in (0, 1)$, such that:*

$$\Pr_{q \sim P_\mathcal{K}} \left( d(q, \mathcal{S}_y) \leq R_\mathcal{S}, \ \|q - \mu_y\|_2 \leq R_\mu \right) \geq 1 - \alpha \tag{18}$$

$$\Pr_{q' \sim P_\mathcal{U}} \left( m_\mathcal{S}(q') \geq \Delta_\mathcal{S}, \ m_\mu(q') \geq \Delta_\mu \right) \geq 1 - \beta \tag{19}$$

*If the probabilistic margin condition $\Delta_\mathcal{S}^2 + \lambda \Delta_\mu^2 > R_\mathcal{S}^2 + \lambda R_\mu^2$ holds, then for any threshold $\delta$ satisfying*

$$R_\mathcal{S}^2 + \lambda R_\mu^2 < \delta < \Delta_\mathcal{S}^2 + \lambda \Delta_\mu^2 \tag{20}$$

*the unified score $s(\cdot)$ satisfies $\Pr_{q \sim P_\mathcal{K}}(s(q) > \delta) \leq \alpha$ and $\Pr_{q' \sim P_\mathcal{U}}(s(q') \leq \delta) \leq \beta$. Consequently, under a single threshold $\delta$, the false rejection rate on known queries is at most $\alpha$, and the false acceptance rate on unknown queries is at most $\beta$.*

*Proof.* For known queries, define the event $A_\mathcal{K} := \{d(q, \mathcal{S}_y) \leq R_\mathcal{S}, \ \|q - \mu_y\|_2 \leq R_\mu\}$.

By assumption Eq. (18), $\Pr(A_\mathcal{K}) \geq 1 - \alpha$. On $A_\mathcal{K}$, we have $D_y(q) = d(q, \mathcal{S}_y)^2 + \lambda \|q - \mu_y\|_2^2 \leq R_\mathcal{S}^2 + \lambda R_\mu^2$.

Since $s(q) = \min_i D_i(q) \leq D_y(q)$, it follows that $s(q) \leq R_\mathcal{S}^2 + \lambda R_\mu^2 < \delta$, where the last inequality follows from Eq. (20).

the event $\{s(q) > \delta\}$ is contained in $A_\mathcal{K}^c$ and Therefore

$$\Pr_{q \sim P_\mathcal{K}}(s(q) > \delta) \leq \Pr(A_\mathcal{K}^c) \leq \alpha \tag{21}$$

For unknown queries, define $A_\mathcal{U} := \{m_\mathcal{S}(q') \geq \Delta_\mathcal{S}, \ m_\mu(q') \geq \Delta_\mu\}$.

By assumption Eq. (19), $\Pr(A_\mathcal{U}) \geq 1 - \beta$. On $A_\mathcal{U}$, for every $i \leq T_{\text{seen}}$, $d(q', \mathcal{S}_i) \geq \Delta_\mathcal{S}$ and $\|q' - \mu_i\|_2 \geq \Delta_\mu$. Thus, for every $i$, $D_i(q') = d(q', \mathcal{S}_i)^2 + \lambda \|q' - \mu_i\|_2^2 \geq \Delta_\mathcal{S}^2 + \lambda \Delta_\mu^2$.

Taking the minimum over $i$, we obtain $s(q') = \min_i D_i(q') \geq \Delta_\mathcal{S}^2 + \lambda \Delta_\mu^2 > \delta$,

where the last inequality again follows from Eq. (20). Hence, the event $\{s(q') \leq \delta\}$ can only occur if $A_\mathcal{U}^c$ occurs. Therefore

$$\Pr_{q' \sim P_\mathcal{U}}(s(q') \leq \delta) \leq \Pr(A_\mathcal{U}^c) \leq \beta. \tag{22}$$

which completes the proof. $\qquad\square$

**Proposition B.3** (Routing Margin). *Let $q$ be a query belonging to the true task $y$, satisfying the intra-task compactness bounds $d(q, \mathcal{S}_y) \leq R_\mathcal{S}$ and $\|q - \mu_y\|_2 \leq R_\mu$. Define the minimum inter-task separability as $D_\mathcal{S} \triangleq \min_{c \neq y} d(\mathcal{S}_y, \mathcal{S}_c)$ and $D_\mu \triangleq \min_{c \neq y} \|\mu_y - \mu_c\|_2$. Then, for any incorrect task $c \neq y$, the energy gap is lower-bounded by:*

$$D_c(q) - D_y(q) \geq \left[ (D_\mathcal{S} - R_\mathcal{S})^2 - R_\mathcal{S}^2 \right] + \lambda \left[ (D_\mu - R_\mu)^2 - R_\mu^2 \right]. \tag{23}$$

---

**Algorithm 1** Training Procedure of SAFER

---

**Require:** Task stream $\{\mathcal{T}^t\}_{t=1}^T$ with corresponding graph $\{\mathcal{G}^t\}_{t=1}^T$; Backbone $f_\theta$; Hyperparameters.
**Ensure:** Learned parameters $\{P_t, W_t, \phi_t, g_t\}$ and Fingerprints $\{\mathcal{S}_t\}$.
 1: **Initialization:** Pre-trained GNN encoder $f_\theta$ on $\mathcal{G}^1$ and **freeze** it.
 2: **for** $t = 1$ to $T$ **do**
 3:     Initialize task-specific parameters: $P_t, W_t, \phi_t, g_t$.
 4:     **for** each epoch **do**
 5:         **for** sampled mini-batch from $\mathcal{G}^t$ **do**
 6:             *// 1. Input-Level Adaptation*
 7:             Compute diffused features $X^{\mathrm{ppr}}$ via Eq. (1).
 8:             Compute prompt-injected representations $\tilde{X}$ via Eq. (15).
 9:             *// 2. Optimization with Regularization*
10:             Extract representation $Z = f_\theta(\tilde{X})$.
11:             Update $\{P_t, W_t, \phi_t, g_t\}$ to minimize $\mathcal{L}_{\mathrm{total}}$ as Eq. (17).
12:         **end for**
13:     **end for**
14:     *// 3. Drift-resilient Fingerprint*
15:     Compute centroid $\mu_t$ and subspace basis $U_t$ on the learned features.
16:     Store task fingerprint $\mathcal{S}_t = (\mu_t, U_t)$.
17: **end for**

---

*Proof.* Recall that $D_t(q) \triangleq d(q, \mathcal{S}_t)^2 + \lambda \|q - \mu_t\|_2^2$.

For the true task $y$, the assumptions $d(q, \mathcal{S}_y) \leq R_{\mathcal{S}}$ and $\|q - \mu_y\|_2 \leq R_\mu$ imply $D_y(q) = d(q, \mathcal{S}_y)^2 + \lambda \|q - \mu_y\|_2^2 \leq R_{\mathcal{S}}^2 + \lambda R_\mu^2$.

Now fix any incorrect task $c \neq y$. By definition $D_{\mathcal{S}} = \min_{c \neq y} d(\mathcal{S}_y, \mathcal{S}_c)$, we have $d(\mathcal{S}_y, \mathcal{S}_c) \geq D_{\mathcal{S}}$. For any $u \in \mathcal{S}_y$ and any $v \in \mathcal{S}_c$, the reverse triangle inequality gives $\|q - v\|_2 \geq \big|\|u - v\|_2 - \|q - u\|_2\big| \geq \|u - v\|_2 - \|q - u\|_2$.

Taking $\inf_{v \in \mathcal{S}_c}$ on the left-hand side and on the first term on the right-hand side yields $d(q, \mathcal{S}_c) = \inf_{v \in \mathcal{S}_c} \|q - v\|_2 \geq d(u, \mathcal{S}_c) - \|q - u\|_2$.

Taking $\inf_{u \in \mathcal{S}_y}$ on the right-hand side, and using $\inf_{u \in \mathcal{S}_y} d(u, \mathcal{S}_c) = d(\mathcal{S}_y, \mathcal{S}_c)$ and $\inf_{u \in \mathcal{S}_y} \|q - u\|_2 = d(q, \mathcal{S}_y)$, we obtain $d(q, \mathcal{S}_c) \geq d(\mathcal{S}_y, \mathcal{S}_c) - d(q, \mathcal{S}_y) \geq D_{\mathcal{S}} - R_{\mathcal{S}}$, and hence

$$d(q, \mathcal{S}_c)^2 \geq (D_{\mathcal{S}} - R_{\mathcal{S}})^2. \tag{24}$$

Similarly, by the reverse triangle inequality, $\|q - \mu_c\|_2 = \|(q - \mu_y) + (\mu_y - \mu_c)\|_2 \geq \|\mu_y - \mu_c\|_2 - \|q - \mu_y\|_2$. By definition $D_\mu = \min_{c \neq y} \|\mu_y - \mu_c\|_2$, we have $\|\mu_y - \mu_c\|_2 \geq D_\mu$, and together with $\|q - \mu_y\|_2 \leq R_\mu$, $\|q - \mu_c\|_2 \geq D_\mu - R_\mu$, so

$$\lambda \|q - \mu_c\|_2^2 \geq \lambda (D_\mu - R_\mu)^2. \tag{25}$$

Combining (24) and (25) yields $D_c(q) = d(q, \mathcal{S}_c)^2 + \lambda \|q - \mu_c\|_2^2 \geq (D_{\mathcal{S}} - R_{\mathcal{S}})^2 + \lambda (D_\mu - R_\mu)^2$. Therefore, $D_c(q) - D_y(q) \geq \big((D_{\mathcal{S}} - R_{\mathcal{S}})^2 - R_{\mathcal{S}}^2\big) + \lambda \big((D_\mu - R_\mu)^2 - R_\mu^2\big)$, which proves the claim. $\qquad\square$

## C. Algorithm.

The training and inference are summarized in Alg. 1 and Alg. 2, respectively.

## D. Experimental setup.

### D.1. More Implementation Details.

All the continual learning methods, including the proposed method, are implemented based on the GCLB (Zhang et al., 2022a). We employ a single-layer GCN model as the backbone. Specifically, the hidden dimension is set to 256 for all

---

**Algorithm 2** Inference of SAFER

---

**Require:** Query $x$; Frozen Anchor $f_\theta$; Fingerprints $\{\mathcal{S}_t\}$; Threshold $\delta$.
**Ensure:** Prediction $\hat{y}$.
1: // *1. Geometric Routing:*
2: Map query to anchor space: $z \leftarrow f_\theta(x^{\text{ppr}})$.
3: Identify target task $\hat{\tau}$ and uncertainty $s$:
4:    $\hat{\tau} = \arg\min_t D_t(z), \quad s = D_{\hat{\tau}}(z). \quad$ // *via Eq.* (8)
5: // *2. Task-Specific Prediction:*
6: Adapt input using prompts of task $\hat{\tau}$:
7:    $\tilde{x} \leftarrow x^{\text{ppr}} + \text{softmax}(x^{\text{ppr}}W_{\hat{\tau}})P_{\hat{\tau}}$.
8: Compute candidate class: $\hat{y} = \arg\max g_{\hat{\tau}}(f_\theta(\tilde{x}))$.
9: // *3. Open-Set Decision:*
10:    $\hat{y} = \begin{cases} \text{Unknown,} & \text{if } s > \delta \\ \hat{y}, & \text{otherwise} \end{cases}$

---

methods. The number of training epochs of each graph learning task is 200, with Adam as the optimizer, and the learning rate is set to 0.005 by default.

**Pre-training for GNN Encoder.** Following the setup in TPP (Niu et al., 2024b), we adopt an unsupervised pre-trained GNN as the encoder and keep its parameters frozen, while still allowing gradients to backpropagate through it. Specifically, from the first task $\mathcal{G}^1$, two widely used graph augmentations are employed, i.e., edge removal and attribute masking to obtain the corrupted graph $\tilde{\mathcal{G}}^1$, then both $G^1$ and $\tilde{\mathcal{G}}^1$ are inputted into the shared GNNs $f_\theta(\cdot)$ to obtain the node embeddings $Z^1$ and $\tilde{Z}^1$. Through graph contrastive learning, we try to enhance the semantic similarity between the same node embeddings in different views, as well as moving away other node embeddings. The pairwise objective for each node pair $(z_i^1, \tilde{z}_i^1)$ can be formulated as:

$$\ell(\tilde{z}_i^1, z_i^1) = -\log \frac{\exp\left(\text{sim}(\tilde{z}_i^1, z_i^1)/\tau\right)}{\exp\left(\text{sim}(\tilde{z}_i^1, z_i^1)/\tau\right) + \sum_{j \neq i}^N \exp\left(\text{sim}(\tilde{z}_i^1, z_j^1)/\tau\right) + \sum_{j \neq i}^N \exp\left(\text{sim}(\tilde{z}_i^1, \tilde{z}_j^1)/\tau\right)} \tag{26}$$

where $sim(\cdot)$ represents the cosine similarity and $\tau$ is a temperature hyperparameter. And the overall objective can be defined as follows:

$$\ell_{contrast} = \frac{1}{2N} \sum_{i=1}^N (\ell(\tilde{z}_i^1, z_i^1) + \ell(z_i^1, \tilde{z}_i^1)) \tag{27}$$

For graph contrastive learning, the probabilities of edge removal and attribute masking are set to 0.2 and 0.3 respectively for all datasets. Besides, the learning rate is set to 0.001 with Adam optimizer, the training epochs are set to 200 and the temperature $\tau$ is 0.5 for all datasets.

The code is implemented with PyTorch (version: 1.13.1), DGL (version: 1.0.1), OGB (version: 1.3.6), and Python 3.8.20. Besides, all experiments are conducted on a Linux server with an Intel CPU (Intel Xeon Platinum 8352V) and a NVIDIA GPU (RTX 4090).

### D.2. Details on Datasets.

Following CGLB (Zhang et al., 2022a), we use four large-scale GCIL datasets in our experiments:

- **CoraFull** (McCallum et al., 2000): a citation network with 70 classes, where nodes represent papers and edges denote citation links.

- **Arxiv** (Hu et al., 2020): a citation network of Computer Science arXiv papers indexed by MAG (Sinha et al., 2015). Each node denotes a CS paper, and edges represent citations. Nodes are classified into 40 subject areas. Node features are computed as the average word embedding of words in the title and abstract.

- **Reddit** (Hamilton et al., 2017): a graph of Reddit posts from September 2014, where each post is labeled by its

*Table 6.* Key statistics of the graph datasets.

| Datasets | CoraFull | Arxiv | Reddit | Products |
|---|---|---|---|---|
| # nodes | 19,793 | 169,343 | 227,853 | 2,449,028 |
| # edges | 130,622 | 1,166,243 | 114,615,892 | 61,859,036 |
| # classes | 70 | 40 | 40 | 46 |
| # tasks | 35 | 20 | 20 | 23 |
| # Avg. nodes per task | 660 | 8,467 | 11,393 | 122,451 |
| # Avg. edges per task | 4,354 | 58,312 | 5,730,794 | 2,689,523 |

community (subreddit). Nodes represent posts, and an edge exists if a user has commented on both posts. Node features are derived from attributes including title, content, comments, post score, and the number of comments.

- **Products** (Hu et al., 2020): an Amazon co-purchasing network, where nodes represent products and edges indicate products that are frequently bought together. Node features are constructed from dimensionality-reduced bag-of-words of product descriptions.

The statistics of the used graph datasets are summarized in Table 6.

### D.3. Descriptions of Baselines.

Since most GCIL baselines are not designed for open-set evaluation, we adopt a unified protocol and additionally compute a consistent energy-style unknownness score for all methods:

$$E_e(x) = -T \cdot log \sum_c exp(\frac{logit_c}{T}) \tag{28}$$

where T is the temperature hyperparameter. Moreover, for TPP we additionally introduce a prototype-based knownness score (DTPP) to improve open-set separation:

$$E_{dist}(x) = -\min_{t \in \{0,...,k\}} \|z(x) - p_t\|_2, \tag{29}$$

where $z(x)$ is the embedding of $x$ and $p_t$ denotes the prototype of task $t$ constructed from seen tasks. Intuitively, a larger $s(x)$ indicates that $x$ is more likely to be *known*. Finally, we introduce a Mix scoring strategy (MTPP) that combines the energy score and the prototype-distance score for evaluation:

$$E_{mix}(x) = \lambda E_e(x) + E_{dist}(x). \tag{30}$$

- **EWC** (Kirkpatrick et al., 2017) is a regularization-based method that adds a quadratic penalty on model parameters according to their importance to previous tasks, thereby maintaining performance on past tasks.

- **MAS** (Aljundi et al., 2018) preserves parameters that are important to previous tasks based on the sensitivity of predictions to parameter changes.

- **GEM** (Lopez-Paz & Ranzato, 2017) stores representative samples in an episodic memory and modifies gradients of the current task using gradients computed on the memory to mitigate forgetting.

- **LwF** (Li & Hoiem, 2017) employs knowledge distillation to minimize the discrepancy between logits of the old and new models, preserving previous knowledge.

- **TWP** (Liu et al., 2021) regularizes important parameters in topological aggregation and loss optimization to retain performance on prior tasks.

- **ERGNN** (Zhou & Cao, 2021) is a replay-based method that constructs memory by storing representative nodes selected from previous tasks.

- **SSM** (Zhang et al., 2022b) incorporates explicit topology of selected nodes by storing sparsified computation subgraphs as memory for graph continual learning.

- **SEM** (Zhang et al., 2023b) improves SSM by storing the most informative topology via Ricci curvature-based graph sparsification.

- **CaT** (Liu et al., 2023) condenses each task graph into a small synthesized replay graph and maintains a condensed graph memory; continual learning is performed by updating the model directly on this condensed memory.

- **DeLoMe** (Niu et al., 2024a) learns lossless prototypical node representations as memory to capture holistic information of previous tasks, and further adopts a debiased GCL loss to address class imbalance between memory and current data.

- **DMSG** (Qiao et al., 2024) includes a holistic and efficient buffer selection module and a generative memory replay module to effectively prevent the model from forgetting previous tasks when learning new tasks.

- **PUMA** (Liu et al., 2024)leverages graph condensation to compress historical graphs into compact synthetic graphs, reducing memory overhead while mitigating catastrophic forgetting.

- **TPP** (Niu et al., 2024b) transductively captures task-specific prototypes utilizing a Laplacian smoothing-based matching approach, achieving 100% task-ID prediction accuracy and 0% forgetting ratio.

### D.4. Evaluation Metrics.

**Average Performance (AP) and Average Forgetting (AF).** AP and AF are computed from the lower-triangular accuracy matrix $\mathbf{M} \in \mathbb{R}^{T \times T}$, where $T$ is the number of tasks. The entry $M_{t,j}$ $(t \geq j)$ denotes the classification accuracy on task $j$ after the model has been optimized on task $t$. Therefore, the $t$-th row $\mathbf{M}_{t,:}$ records the performance on all previously learned tasks after learning task $t$, while the $j$-th column $\mathbf{M}_{:,j}$ characterizes the performance trajectory on task $j$ as different tasks are learned. After learning all $T$ tasks, the overall average accuracy (AP) and average forgetting (AF) are computed as:

$$\text{AP} = \frac{1}{T} \sum_{j=1}^{T} M_{T,j}, \qquad \text{AF} = \frac{1}{T-1} \sum_{j=1}^{T-1} \left( M_{T,j} - M_{j,j} \right). \tag{31}$$

**Area under the ROC curve.** The ROC curve presents a trade-off between the true positive rate and the false positive rate under different thresholds in ranging from 0 to 1. Intuitively, the AUROC can be understood as the probability that for any pair of a positive example (i.e., known sample) and a negative example (unknown sample), the positive one has a larger estimated score than the negative.

**Open-set classification rate.** We split the test samples into samples from known classes $\mathcal{D}_c$ and samples from unknown classes $\mathcal{D}_u$. Let $\theta$ be a score threshold. For samples from $\mathcal{D}_c$, we calculate the Correct Classification Rate (CCR) as the fraction of the samples where the correct class $\hat{c}$ has maximum probability and has a probability greater than $\theta$. We compute the False Positive Rate (FPR) as the fraction of samples from $\mathcal{D}_u$ that are classified as any known class $c \in \mathcal{C}$ with a probability greater than $\theta$.

$$\text{FPR}(\theta) = \frac{|\{x \mid x \in \mathcal{D}_u \wedge \max_c P(c \mid x) \geq \theta\}|}{|\mathcal{D}_u|}, \tag{32}$$

$$\text{CCR}(\theta) = \frac{|\{x \mid x \in \mathcal{D}_c \wedge \arg\max_c P(c \mid x) = \hat{c} \wedge P(\hat{c} \mid x) > \theta\}|}{|\mathcal{D}_c|}. \tag{33}$$

Finally, we plot CCR versus FPR, varying the probability threshold from large $\theta$ on the left side to small $\theta$ on the right side. For the smallest $\theta$, the CCR is identical to the closed-set classification accuracy on $\mathcal{D}_c$. Unlike the above discussed evaluation measures, which are prone to dataset bias, OSCR is not since its DIR axis is computed solely from samples belonging to $\mathcal{D}_c$. Moreover, when algorithms exposed to different numbers of samples from $\mathcal{D}_a$ need to be compared, rather than using the normalized FPR with an algorithm-specific $\mathcal{D}_a$, we may use the raw number of false positives on the horizontal axis.

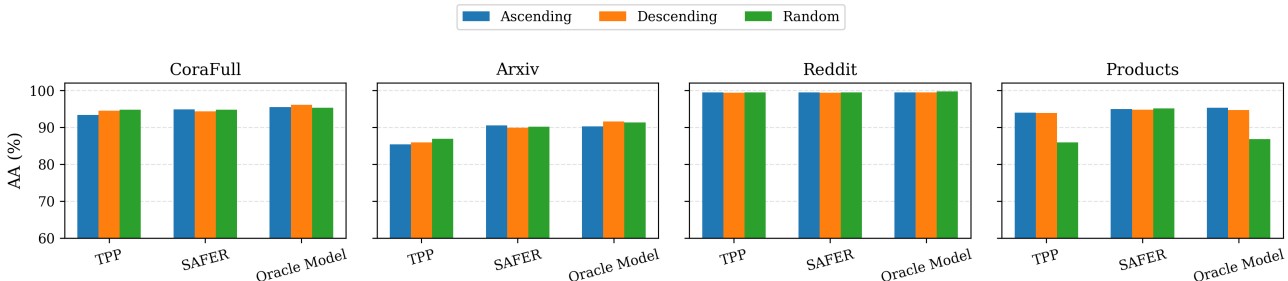

*Figure 8.* Performance under different task orders.

*Table 7.* **Results under different open-set task-order settings.** We compare DMSG, TPP, and SAFER under similarity-based task order, size-ascending task order, and permanently unknown classes. "↑" indicates that higher is better.

| Setting | Method | CoraFull | | | | Arxiv | | | | Reddit | | | | Products | | | |
|---|---|---|---|---|---|---|---|---|---|---|---|---|---|---|---|---|---|
| | | AP↑ | AF↑ | AUC↑ | OSCR↑ | AP↑ | AF↑ | AUC↑ | OSCR↑ | AP↑ | AF↑ | AUC↑ | OSCR↑ | AP↑ | AF↑ | AUC↑ | OSCR↑ |
| Similarity | DMSG | 77.8 | 1.6 | 66.2 | 62.4 | 47.6 | -5.4 | 53.7 | 37.4 | 81.1 | -17.2 | 56.0 | 54.5 | 54.0 | -14.2 | 61.7 | 36.9 |
| | TPP | 91.6 | 0.0 | 80.9 | 79.9 | 85.4 | 0.0 | 58.8 | 57.7 | 99.4 | 0.0 | 71.0 | 70.9 | 87.2 | 0.0 | 69.1 | 68.5 |
| | **SAFER** | **93.5** | **0.0** | **97.0** | **93.5** | **90.0** | **0.0** | **100.0** | **91.4** | **99.4** | **0.0** | **100.0** | **99.7** | **94.6** | **0.0** | **100.0** | **97.8** |
| Size-asc. | DMSG | 78.2 | 2.1 | 66.6 | 62.5 | 48.1 | -5.0 | 54.1 | 37.7 | 85.8 | -10.9 | 55.4 | 53.2 | 54.7 | -11.4 | 61.9 | 38.4 |
| | TPP | 91.6 | 0.0 | 80.9 | 79.9 | 85.4 | 0.0 | 58.8 | 57.7 | 99.4 | 0.0 | 71.0 | 70.9 | 87.2 | 0.0 | 69.1 | 68.6 |
| | **SAFER** | **93.5** | **0.0** | **97.0** | **93.6** | **90.0** | **0.0** | **100.0** | **91.7** | **99.4** | **0.0** | **100.0** | **99.7** | **95.6** | **0.0** | **100.0** | **97.8** |
| Perm-unk. | DMSG | 79.5 | 1.5 | 82.4 | 75.2 | 48.4 | -3.8 | 53.3 | 37.7 | 94.6 | -2.3 | 56.5 | 56.1 | 54.6 | -11.7 | 47.5 | 33.2 |
| | TPP | 94.3 | 0.0 | 82.2 | 81.0 | 84.3 | 0.0 | 63.9 | 62.6 | 99.5 | 0.0 | 73.6 | 73.5 | 90.7 | 0.0 | 91.0 | 90.0 |
| | **SAFER** | **94.5** | **0.0** | **100.0** | **95.7** | **93.3** | **0.0** | **100.0** | **92.6** | **98.6** | **0.0** | **100.0** | **99.0** | **95.2** | **0.0** | **100.0** | **97.9** |

# E. Extended Experiments.

## E.1. Robustness to Different Task Orders.

We assign two node classes to each task and follow the standard protocol in CGLB for fair comparisons. Concretely, for a multi-class graph dataset, we partition classes into tasks by the ascending order of original labels (e.g., 0,1, 2,3, 4,5, etc.). To assess robustness to task formulation, we also construct tasks using two alternative splits: descending label order and random class ordering. As shown in Figure 8, our method remains robust across different task orders, achieving performance comparable to TPP, as well as narrowing the gap with the Oracle model.

## E.2. Robustness under different open-set task-order settings.

Table 7 evaluates the robustness of DMSG, TPP, and SAFER under three open-set task-order settings. In the similarity-based setting, after each task, the next training task is selected as the one most similar to the current task. In the size-ascending setting, tasks are learned sequentially according to the amount of data contained in each task. In the permanently unknown setting, the last two tasks are never used for training; instead, after learning each incremental task, samples from these two tasks are always treated as unknown classes for open-set detection, rather than using only the next task as the unknown task. SAFER consistently achieves the strongest overall performance across the four datasets and all task-order settings, with particularly large gains on AUC and OSCR. Compared with TPP, SAFER preserves competitive classification performance while substantially improving open-set discrimination, indicating that the proposed routing scores can more reliably separate known and unknown samples under varying task sequences. The improvements remain stable even in the permanently unknown setting, suggesting that SAFER is less sensitive to task-order changes and can better generalize to unseen classes that never appear during incremental training.

## E.3. Results with Different GNN Backbones

We extended our experiments by incorporating SGC and GAT as backbones, and we compared the performance of SAFER on all four datasets. The results are summarized in Table 8.

*Table 8.* Backbone comparison results on four datasets.

| Setting | CoraFull | | | | Arxiv | | | | Reddit | | | | Products | | | |
|---|---|---|---|---|---|---|---|---|---|---|---|---|---|---|---|---|
| | AP | AF | AUC | OSCR | AP | AF | AUC | OSCR | AP | AF | AUC | OSCR | AP | AF | AUC | OSCR |
| GCN | 94.9 | 0.0 | 97.9 | 94.6 | 90.6 | 0.0 | 100.0 | 92.4 | 99.5 | 0.0 | 100.0 | 99.6 | 95.0 | 0.0 | 100.0 | 97.1 |
| SGC | 92.7 | 0.0 | 97.1 | 94.1 | 90.4 | 0.0 | 100.0 | 91.7 | 99.5 | 0.0 | 100.0 | 99.7 | 95.7 | 0.0 | 100.0 | 97.7 |
| GAT | 94.6 | 0.0 | 99.2 | 95.8 | 91.2 | 0.0 | 100.0 | 91.8 | 99.5 | 0.0 | 100.0 | 99.7 | 94.9 | 0.0 | 100.0 | 95.8 |
| SGC w/o PPR | 94.9 | 0.0 | 96.2 | 93.8 | 93.2 | 0.0 | 99.9 | 91.7 | 99.5 | 0.0 | 100.0 | 99.7 | 95.7 | 0.0 | 100.0 | 97.6 |

*Table 9.* Robustness under feature noise on four datasets.

| Noise | CoraFull | | | Arxiv | | | Reddit | | | Products | | |
|---|---|---|---|---|---|---|---|---|---|---|---|---|
| | AP | OSCR | Id_ACC | AP | OSCR | Id_ACC | AP | OSCR | Id_ACC | AP | OSCR | Id_ACC |
| 0.00 | 94.9 | 94.6 | 100.0 | 90.6 | 92.4 | 100.0 | 99.4 | 99.6 | 100.0 | 95.0 | 97.1 | 100.0 |
| 0.01 | 95.7 | 91.6 | 100.0 | 93.2 | 90.2 | 100.0 | 99.1 | 99.5 | 100.0 | 94.5 | 96.8 | 100.0 |
| 0.03 | 96.0 | 92.2 | 100.0 | 92.5 | 90.2 | 100.0 | 98.6 | 99.1 | 100.0 | 94.8 | 96.3 | 100.0 |
| 0.05 | 94.6 | 91.2 | 100.0 | 91.8 | 89.5 | 100.0 | 98.1 | 98.8 | 100.0 | 93.0 | 95.8 | 100.0 |
| 0.10 | 93.7 | 90.2 | 100.0 | 90.1 | 88.0 | 100.0 | 97.5 | 98.3 | 100.0 | 92.2 | 94.6 | 100.0 |
| 0.30 | 88.2 | 82.2 | 100.0 | 84.4 | 82.1 | 100.0 | 95.9 | 97.1 | 100.0 | 87.1 | 90.6 | 100.0 |

In our ablation experiments, we observe that PPR is significantly dependent on the backbone: when combined with parameterized nonlinear graph neural networks (GCN, GAT), PPR yields substantial performance gains and reaches the SOTA; however, when combined with a linear propagation model (SGC), performance instead degrades, exhibiting a clear over-smoothing phenomenon. Meanwhile, SGC can already achieve near-SOTA results without introducing PPR.

For SGC, the representation is $Z_{SGC} = S^K X W$, where S is a normalized propagation operator. With the spectral decomposition $S = U\Lambda U^\top$, we obtain $S^K = U\Lambda^K U^\top$, which shows that SGC performs a multi-hop low-pass filtering: as $K$ increases, high-frequency components are increasingly attenuated. For PPR, it is also a diffusion-based low-pass operator:

$$\Pi_{PPR} = \alpha(I_n - (1-\alpha)S)^{-1} \tag{34}$$

where $\alpha \in (0, 1]$. Therefore, when PPR is stacked on top of SGC, the overall propagation becomes $Z = \Pi_{PPR}S^K X W$, which pushes the model into an overly strong smoothing regime, causing embeddings from different classes to collapse toward a similar subspace, reducing separability and leading to performance degradation. In contrast, GAT and GCN include learnable transformations and nonlinear modeling($\sigma(\cdot)$) respectively, which help preserve discriminative information during propagation. In this case, PPR mainly enlarges the effective receptive field and provides multi-hop consistency regularization, while adaptively counterbalance the smoothing effect.

### E.4. Robustness of Performance under Noise

We include additional experiments where each task's graph samples are perturbed with random noise to evaluate **SAFER**'s performance and task-ID prediction accuracy in Table 9. Specifically, we randomly add or remove edges from graphs while fully adhering to the GCIL setting. As the noise level increases, performance decreases moderately across the four datasets, which is expected. More importantly, we find that the accuracy of task-ID prediction is always 100% and for small-scale datasets, introducing a certain amount of noise can even improve the model's performance. When faced with open-set conditions and noise, our subspace fingerprint module can robustly perform routing and reject unknown tasks.

### E.5. Parameter Study

**Sensitivity Analysis of PPR propagation Step and Prompt Number.** To study the sensitivity of the number of PPR propagation and prompt. We report AP and OSCR for each setting in Figure 9. The result indicates that SAFER is robust to

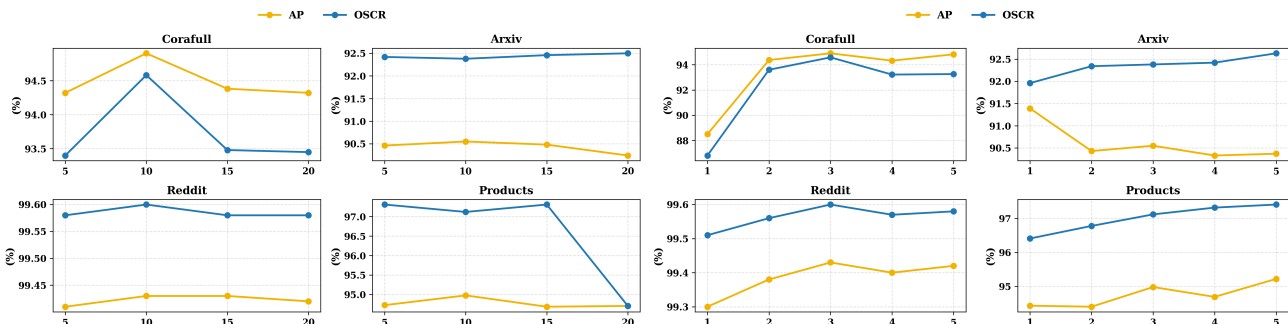

*Figure 9.* The sensitivity of the PPR propagation step (left) and the prompt number (right).

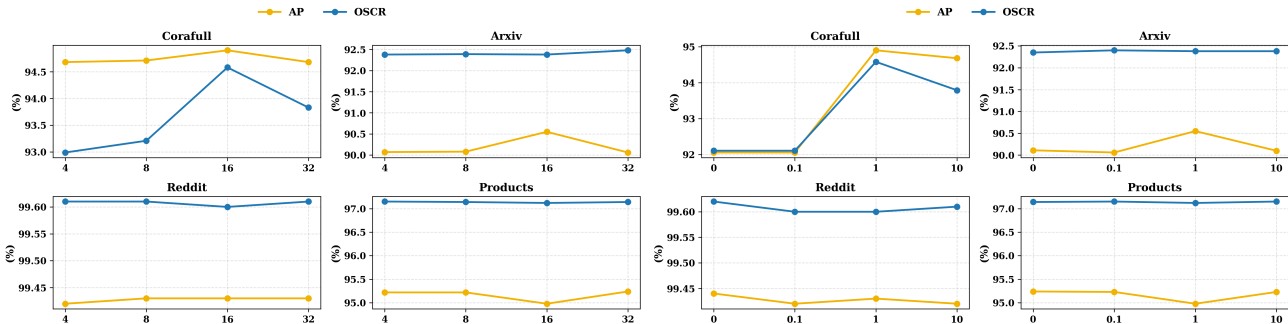

*Figure 10.* The sensitivity of PCA rank (left) and $\lambda$ (right).

*Table 10.* Efficiency comparison in inference time and peak GPU memory.

| Method | CoraFull | | Arxiv | | Reddit | | Products | |
|--------|-----------|-------------|-----------|-------------|-----------|-------------|-----------|-------------|
| | Inference (s) | Memory (MB) | Inference (s) | Memory (MB) | Inference (s) | Memory (MB) | Inference (s) | Memory (MB) |
| DMSG | 11.4 | 7675.4 | 7.5 | 1386.3 | 45.9 | 3214.4 | 873.2 | 8014.3 |
| TPP | 12.5 | 262.6 | 2.0 | 230.0 | 33.5 | 2454.1 | 89.1 | 7732.0 |
| SAFER | 16.3 | 469.7 | 3.1 | 213.1 | 43.7 | 2402.3 | 105.8 | 10532.0 |

a reasonable selection of PPR propagation step and prompt number, while careful tuning can not be ignored.

**Sensitivity Analysis of PCA rank and $\lambda$.** To study the sensitivity of the PCA rank and $\lambda$ in Eq. (7), we vary them within $\{4, 8, 16, 32\}$ and $\{0, 0.1, 1, 10\}$ respectively. We find that the performance of SAFER contains a reasonable range across different datasets when rank changes in Figure 10. Besides, for a specific dataset, the second term plays an important role in avoiding degenerate matches when relying on the subspace alone.

## F. Time Complexity Analysis.

We analyze the time complexity of SAFER by dividing it into four main components:

- **Structure-aware signal enhancement:** $\mathcal{O}\big(K\,(|E|d + Nd) + |E|d + Nd^2\big)$, where $N$ and $|E|$ denote the numbers of nodes and edges, $d$ is the feature dimension, $K$ is the number of PPR diffusion steps. The $K(|E|d + Nd)$ term accounts for PPR diffusion, while $|E|d + Nd^2$ accounts for the GCN forward pass. With graph-level caching, the PPR term is computed once per graph object and becomes amortized.

- **Geometric subspace fingerprints:** $\mathcal{O}\big(|E|d + Nd^2 + Md^2\big)$, where $M$ is the number of training nodes used to form the fingerprint. And $Md^2$ reflects the SVD on an $M \times h$ embedding matrix.

- **Adapter-mediated geometric regularization:** $\mathcal{O}\big(N(d^2 + dp) + B^2p\big)$, where $p$ is the projection dimension of the

adapter, and $B$ is the number of samples used in supervised contrastive learning; the term $N(d^2 + dp)$ corresponds to the two-layer MLP projection over node embeddings, while $B^2 p$ accounts for the pairwise similarity computation in SupCon.

- **Task-adaptive module:** $\mathcal{O}(Nd^2 P + NdC)$, where $P$ is the prompt pool size and $C$ is the number of classes per task; the term $Nd^2 P$ accounts for attention-based prompt construction, and $NdC$ is the cost of the linear classifier head.

SAFER not only performs classification over known classes at inference time, but also supports open-set recognition by rejecting unknown classes. In contrast, DMSG and TPP follow a standard closed-set setting and only predict within a fixed label space. As shown in Table 10, despite the more demanding objective, SAFER still remains in the same order of magnitude as the closed-set baselines, which indicates SAFER achieves efficient and resource-controlled inference.

