# OpenReview forum: "Subspace-Aware Feature Reshaping for Open-Set Graph Class-Incremental Learning"
_ICML.cc/2026/Conference — ICML 2026 regular_

### Official Review · Reviewer_HJuw · 2026-02-23

**Soundness:** 3
**Presentation:** 4
**Significance:** 3
**Originality:** 4
**Overall Recommendation:** 5
**Confidence:** 4

**Summary:**

This paper addresses a highly practical and challenging problem in Graph Class-Incremental Learning by introducing the Open-Set scenario. Existing GCIL methods typically assume a closed-set test distribution, which often leads to severe representation drift and overconfidence when encountering unknown classes in real-world evolving graphs. To tackle this, the authors propose a novel framework named SAFER, which leverages task-adaptive subspace expansion, prompt-driven graph structural transformation, and margin-based orthogonal subspace contrastive learning. This design not only mitigates catastrophic forgetting by isolating task-specific features but also naturally reserves an "open space" to reject unknown classes. The paper is well-motivated, the methodology is intuitively sound, and the extensive experiments on various benchmark datasets effectively demonstrate the superiority of the proposed framework.

**Compliance With Llm Reviewing Policy:**

Affirmed.

**Final Justification:**

This work focuses on open-set graph class-Incremental learning. The proposed SAFER framework addresses the issues of catastrophic forgetting and unknown-class detection through orthogonal subspaces and feature reshaping. During the initial review, I found the paper's motivation to be clear and its methodological design to be sound.
During the review stage, I had two main concerns: first, the generality of the framework’s GNN backbone, and second, the approach to topological handling under the incremental setting. The authors provided clear responses to both issues. Considering the practical significance of the research motivation and the originality of the methodology, I maintain my recommendation for acceptance (5).

**Key Questions For Authors:**

Please see the weaknesses.

**Limitations:**

Yes.

**Strengths And Weaknesses:**

Strengths:
1. The integration of Open-Set Recognition with Graph Class-Incremental Learning accurately reflects the open-ended nature of real-world graph streams, making the research motivation highly compelling.
2. The framework design is highly logical. Using orthogonal subspaces to isolate knowledge effectively prevents catastrophic forgetting, while the margin-based contrastive learning elegantly constrains the distribution of known classes, naturally leaving room for unknown class detection.
3. The consistent performance gains over various "GCIL + OOD detection" baselines validate the effectiveness of the approach.

Weaknesses:
1. As a general framework featuring "feature reshaping," is SAFER's performance strictly tied to a specific Graph Neural Network backbone? The authors should conduct a simple substitution experiment to see if SAFER consistently outperforms the baselines under different message-passing mechanisms.
2. In an incremental setting, nodes from new tasks will inevitably share edges with nodes from old tasks. Does forcing features from different tasks into strictly orthogonal subspaces disrupt or block this natural cross-task information passing? The authors can add a discussion explaining how the model preserves and balances this topological structural information during the message-passing phase, prior to the orthogonal

---

> ### Author Rebuttal · Authors · 2026-03-30
>
> Thank you for your thoughtful and valuable feedback.
>
> **Weakness #1:**
>
> We emphasize that SAFER is a general framework and is not restricted to a particular GNN backbone. To further validate this point, we conducted **GCN**, **SGC**, and **GAT**.
>
> In our ablation experiments, we observe that PPR is signiﬁcantly dependent on the backbone: when combined with parameterized nonlinear graph neural networks (GCN, GAT), PPR yields substantial performance gains and reaches the SOTA; however, when combined with a linear propagation model (SGC), performance instead degrades, exhibiting a clear over-smoothing phenomenon. Meanwhile, SGC can already achieve near-SOTA results without introducing PPR. You can find more details in appendix E2.
>
> | Dataset     | CoraFull |         |          |          | Arxiv    |         |           |          | Reddit   |         |           |          | Products |         |           |          |
> | :---------- | :------- | :------ | :------- | :------- | :------- | :------ | :-------- | :------- | -------- | ------- | --------- | -------- | -------- | ------- | --------- | -------- |
> | Method      | AA↑      | AF↑     | AUC↑     | OSCR↑    | AA↑      | AF↑     | AUC↑      | OSCR↑    | AA↑      | AF↑     | AUC↑      | OSCR↑    | AA↑      | AF↑     | AUC↑      | OSCR↑    |
> | SGC         | 93.3±0.7 | 0.0±0.0 | 97.3±0.6 | 93.8±0.4 | 90.6±0.2 | 0.0±0.0 | 100.0±0.0 | 92.1±0.2 | 99.5±0.0 | 0.0±0.0 | 100.0±0.0 | 99.7±0.0 | 96.1±0.3 | 0.0±0.0 | 100.0±0.0 | 97.8±0.1 |
> | GCN         | 94.9±0.2 | 0.0±0.0 | 97.9±0.3 | 94.6±0.6 | 90.6±0.1 | 0.0±0.0 | 100.0±0.0 | 92.4±0.2 | 99.5±0.0 | 0.0±0.0 | 100.0±0.0 | 99.6±0.1 | 95.5±0.4 | 0.0±0.0 | 100.0±0.0 | 97.5±0.5 |
> | GAT         | 94.4±0.2 | 0.0±0.0 | 97.2±1.6 | 94.2±1.6 | 90.6±0.5 | 0.0±0.0 | 100.0±0.0 | 91.8±0.0 | 99.5±0.0 | 0.0±0.0 | 100.0±0.0 | 99.7±0.0 | 95.2±0.3 | 0.0±0.0 | 100.0±0.0 | 96.4±0.5 |
> | SGC w/o PPR | 94.7±0.5 | 0.0±0.0 | 96.6±0.4 | 93.6±0.2 | 93.2±0.0 | 0.0±0.0 | 100.0±0.0 | 91.8±0.1 | 99.5±0.0 | 0.0±0.0 | 100.0±0.0 | 99.7±0.0 | 95.9±0.1 | 0.0±0.0 | 100.0±0.0 | 97.7±0.1 |
>
> **Weakness #2:**
>
> In class-incremental continual graph learning, different tasks are defined by partitioning classes. If inter-task edges are retained, the model is allowed to aggregate information from previous tasks through these inter-task edges during the GNN neighborhood aggregation process. Although the labels of previous-task data are inaccessible, information from previous tasks is still inevitably aggregated through inter-task edges.
>
> According to the experimental results reported in the Continual Graph Learning Benchmark, due to the message-passing mechanism of graph neural networks, inter-task edges break the restriction on access to information from previous tasks. This implies that the shortest path between any two nodes is usually very short. Therefore, inter-task edges are likely to propagate a large amount of information from old tasks to new tasks. This effect may alleviate forgetting and benefit performance. In addition, their study shows that the impact of inter-task edges is highly dataset-dependent: inter-task edges introduce conflicting factors that affect performance, and whether the net effect is beneficial or harmful depends strongly on the properties of the dataset.
>
> In our experiments, to prevent access to information from previous tasks through inter-task edges, we adopt the setting without inter-task edges.
>
> **Conclusion**
>
> Thank you again for your constructive suggestions. We hope these results and clarifications resolve all your concerns.
>
> [1] CGLB: Benchmark Tasks for Continual Graph Learning. NeurIPS 2022

---

> > ### Author Rebuttal · Reviewer_HJuw · 2026-03-31
> >
> > The authors solved all of my issues about this manuscript.

---

> > > ### Author Response · Authors · 2026-04-03
> > >
> > > We sincerely appreciate your recognition of our work and the time and effort you have devoted as a reviewer!

---

### Official Review · Reviewer_EXke · 2026-03-11

**Soundness:** 3
**Presentation:** 2
**Significance:** 2
**Originality:** 2
**Overall Recommendation:** 3
**Confidence:** 3

**Summary:**

This paper studies the problem of graph continual learning (GCIL), which aims to address the challenges of learning on dynamically evolving graphs. The authors argue that existing GCIL methods generally assume a closed-set testing distribution, where all test samples originate from previously seen tasks. However, this assumption conflicts with real-world scenarios, where novel and unseen classes may inevitably emerge in the future, forming an open-world setting.

To address this issue, the authors investigate the open-set GCIL problem and propose a new framework named SAFER. SAFER performs subspace-aware feature restructuring through drift-robust fingerprints, unifying task routing and open-set rejection under a single energy-based criterion. In addition, the framework introduces a geometric space consistency regularization, which significantly improves intra-class compactness while mitigating cross-task representation drift.

Extensive experiments on four benchmark datasets demonstrate the strong performance of SAFER.

**Compliance With Llm Reviewing Policy:**

Affirmed.

**Key Questions For Authors:**

See the weaknesses.

**Limitations:**

Yes

**Strengths And Weaknesses:**

**Strengths**

1. The authors not only propose an empirical solution to the challenges in current graph continual learning, but also provide theoretical and empirical analyses to support their approach.

2. The proposed method achieves strong performance across multiple benchmark datasets.

3. In addition, the illustrative figure introduced in Figure 1 helps readers better understand the overall framework and motivation of the paper.

**Weaknesses**

1. Most of the compared baselines are from more than two years ago. The authors should consider including comparisons with more recent methods to better demonstrate the competitiveness of the proposed approach.

2. In Proposition 3.2, the authors assume that the known queries satisfy certain bounded conditions. However, such an assumption may be difficult to guarantee in real-world scenarios, and the authors are encouraged to further justify or discuss the practicality of this assumption.

3. In addition, the source code is not provided, which increases the difficulty of reproducing the proposed method.

---

> ### Author Rebuttal · Authors · 2026-03-30
>
> Thank you for your constructive and detailed review.
>
> **Weakness #1:**
>
> We additionally included two more recent baselines, **PUMA** (TKDE' 2025) and **SERGCL** (LoG' 2024 in Nov.), to provide a more up-to-date comparison. The results show that our method remains competitive against these recent approaches as well.
>
> We also reviewed other recent works on CGL, but most of them are not suitable for direct inclusion in our main comparison table due to fundamental differences in task definitions, evaluation protocols, and dataset settings. We are currently further adapting the corresponding code and experimental setups.
>
> **Table:** Performance on four datasets with different methods.
>
> | Dataset | CoraFull |          |          |          | Arxiv    |           |           |          | Reddit   |          |           |          | Products |          |           |          |
> | :------ | :------- | :------- | :------- | :------- | :------- | :-------- | :-------- | :------- | -------- | -------- | --------- | -------- | -------- | -------- | --------- | -------- |
> | Method  | AA↑      | AF↑      | AUC↑     | OSCR↑    | AA↑      | AF↑       | AUC↑      | OSCR↑    | AA↑      | AF↑      | AUC↑      | OSCR↑    | AA↑      | AF↑      | AUC↑      | OSCR↑    |
> | PUMA    | 73.6±0.9 | -7.0±0.8 | 68.4±0.3 | 50.6±0.3 | 69.3±0.2 | -10.7±0.2 | 61.1±0.3  | 37.4±0.2 | 98.0±0.0 | -0.3±0.0 | 68.9±0.8  | 48.0±0.9 | 74.2±0.4 | -4.1±0.5 | 53.7±0.2  | 35.9±0.2 |
> | SERGCL  | 76.4±0.7 | -8.5±2.0 | 76.3±0.4 | 66.9±0.7 | 66.8±0.5 | -11.6±0.4 | 64.8±0.1  | 51.7±0.3 | 98.6±0.0 | -0.5±0.0 | 75.5±0.5  | 75.5±0.5 | 57.4±0.8 | -6.1±0.5 | 50.7±0.6  | 40.5±0.8 |
> | SAFER   | 94.9±0.2 | 0.0±0.0  | 97.9±0.3 | 94.6±0.6 | 90.6±0.1 | 0.0±0.0   | 100.0±0.0 | 92.4±0.2 | 99.5±0.0 | 0.0±0.0  | 100.0±0.0 | 99.6±0.1 | 95.5±0.4 | 0.0±0.0  | 100.0±0.0 | 97.5±0.5 |
>
> **Weakness #2:**
>
> Proposition 3.2 does not characterize an idealized separation that is assumed to hold naturally in the raw input space. Rather, it specifies the geometric conditions under which the unified energy can induce separation in the learned representation space: known samples should remain sufficiently compact with respect to their true task fingerprints, while unknown samples should stay sufficiently separated from all seen task fingerprints. SAFER supports this geometry by frozen anchor to provide a cross-task comparable reference space, constraining new-task adaptation to lightweight side modules via prompts/adapters, and further improving compactness while suppressing drift through geometric regularization and supervised contrastive learning. Together, these designs shape a more stable and more separable task geometry.
>
> On all four datasets, we additionally measured the empirical geometric gap. We observe that the average minimum $R_\mu$ and $R_S$ of unknown samples with respect to seen task fingerprints are consistently and significantly larger than those of known samples. Moreover, the higher inter/intra separation ratio and the smaller P95 radius of the full model in Fig. 7 further indicate that SAFER indeed pushes the learned representation space toward the geometric regime required by Proposition 3.2.
>
> | **Dataset** | Known $R_{\mu}$ | Known $R_S$ | Unknown $R_{\mu}$ | Unknown $R_S$ | **Avg Gap $R_{\mu}$** | Avg Gap $R_S$ |
> | ----------- | --------------- | ----------- | ----------------- | ------------- | --------------------- | ------------- |
> | CoraFull    | 0.1563          | 0.0874      | 0.5051            | 0.2453        | 0.3429                | 0.1553        |
> | Arxiv       | 0.0425          | 0.0163      | 0.4782            | 0.0988        | 0.4257                | 0.0788        |
> | Reddit      | 0.0109          | 0.0037      | 0.6636            | 0.2471        | 0.6506                | 0.2436        |
> | Products    | 0.1172          | 0.8449      | 0.4487            | 0.4488        | 0.8029                | 0.4216        |
>
> **Weakness #3:**
>
> For reproducibility, our code is available at:  https://anonymous.4open.science/r/SAFER-CB38/,  and the complete implementation will be publicly released on GitHub upon acceptance.
>
> **Conclusion**
>
> Thank you again for your valuable comments. We hope our clarifications and additional analyses have addressed your concerns.
>
> [1] PUMA: Efﬁcient Continual Graph Learning for Node Classiﬁcation with Graph Condensation. TKDE 2025
>
> [2] Stochastic Experience-Replay for Graph Continual Learning. Log 2024

---

> > ### Author Rebuttal · Reviewer_EXke · 2026-04-02
> >
> > Thank you for the detailed response. While I appreciate the additional clarifications, I feel that the theoretical analysis is still not sufficiently integrated with the proposed method, and seems somewhat detached from the core technical contributions. As a result, I will maintain my original score.

---

> > > ### Author Response · Authors · 2026-04-03
> > >
> > > Thank you again for the constructive  feedback.
> > >
> > > Our theoretical analysis is not intended to characterize each training module individually, but rather to provide formal support for the core decision mechanism of our method, the unified residual-energy criterion, and to explain the theoretical feasibility of our method. **The main focus of this work remains the new open-set continual graph learning problem and a competitive model designed for this setting.**
> > >
> > > At the theory level, Proposition 3.1 shows that the residual term has a clear geometric meaning, namely, the distance from a sample to the task affine subspace. Propositions 3.2 and 3.3 further show that when task fingerprints are sufficiently compact and well separated in the representation space, the same unified energy naturally induces an unknown rejection margin and a task routing margin.
> > >
> > > At the method level, the modules in Section 3.5 are designed around these geometric conditions, so as to avoid continuously rewriting the manifolds of historical tasks while improving cross-task separability.
> > >
> > > At the empirical level, after ablating the unified residual-energy router in Fig. 6, unknown detection degrades significantly, indicating that the unified energy criterion itself is the key mechanism that translates the geometric structure of the representation space into routing/rejection gains. Meanwhile, Fig. 7 shows tighter intra-class structure and better inter-class separation, which is consistent with the geometric intuition underlying the theory and further indicates that the full model learns a representation space more favorable for forming a margin.
> > >
> > > To further address your concern, we supplement Proposition 3.2 with a high-probability extension, which relaxes the idealized deterministic assumptions to more practical probabilistic conditions.
> > >
> > > **Proposition 3.2**
> > >
> > > Let $q \sim P_{\mathcal K}$ be a known query with true task $y$, and let $q' \sim P_{\mathcal U}$ be an unknown query. Recall that the task-wise energy is defined as $D_i(x)=d(x,\mathcal S_i)^2+\lambda \lVert x-\mu_i\rVert_2^2$  where $d(x,\mathcal S_i)=\lVert (I-U_iU_i^\top)(x-\mu_i)\rVert_2$  and the unified inference score is $s(x)=\min_{1\le i\le T_{\mathrm{seen}}}D_i(x)$.
> > >
> > > For an unknown query $q'$, define $m_{\mathcal S}(q'):=\min_{1\le i\le T_{\mathrm{seen}}} d(q',\mathcal S_i)$ and $m_{\mu}(q'):=\min_{1\le i\le T_{\mathrm{seen}}}\lVert q'-\mu_i\rVert_2$
> > >
> > > Assume there exist constants $R_{\mathcal S}$, $ R_\mu$ , $ \Delta_{\mathcal S}$, $ \Delta_\mu>0$ and tolerances $\alpha,\beta\in(0,1)$
> > >
> > >
> > >
> > > such that:
> > > $$
> > > \Pr_{q\sim P_{\mathcal K}} \Big( d(q,\mathcal S_y)\le R_{\mathcal S},\ \lVert q-\mu_y\rVert_2\le R_\mu \Big) \ge 1-\alpha
> > > $$
> > >
> > > $$
> > > \Pr_{q'\sim P_{\mathcal U}} \Big( m_{\mathcal S}(q')\ge \Delta_{\mathcal S},\ m_{\mu}(q')\ge \Delta_\mu \Big) \ge 1-\beta
> > > $$
> > >
> > > If the probabilistic margin condition $\Delta_{\mathcal S}^2+\lambda\Delta_\mu^2 > R_{\mathcal S}^2+\lambda R_\mu^2 $ holds, then for any threshold $\delta$ satisfying
> > > $$
> > > R_{\mathcal S}^2+\lambda R_\mu^2 < \delta < \Delta_{\mathcal S}^2+\lambda\Delta_\mu^2
> > > $$
> > > the unified score $s(\cdot)$ satisfies $\Pr_{q\sim P_{\mathcal K}}(s(q)>\delta)\le \alpha$ and $\Pr_{q'\sim P_{\mathcal U}}(s(q')\le \delta)\le \beta$.
> > >
> > > Therefore, under a single threshold $\delta$, the false rejection rate on known queries is at most $\alpha$, and the false acceptance rate on unknown queries is at most $\beta$.
> > >
> > > **Proof**
> > >
> > > For known queries, define the event $A_{\mathcal K} := \{ d(q,\mathcal S_y)\le R_{\mathcal S},\ \lVert q-\mu_y\rVert_2\le R_\mu\}$.
> > >
> > > By assumption $(1)$, $\Pr(A_{\mathcal K})\ge 1-\alpha$.  On $A_{\mathcal K}$ , we have $D_y(q) = d(q,\mathcal S_y)^2+\lambda\lVert q-\mu_y\rVert_2^2 \le R_{\mathcal S}^2+\lambda R_\mu^2. $
> > >
> > > Since $s(q)=\min_{i}D_i(q)\le D_y(q)$, it follows that $s(q)\le R_{\mathcal S}^2+\lambda R_\mu^2<\delta$, where the last inequality follows from $(3)$.
> > >
> > > The event \(\{s(q)>\delta\}\) is contained in \(A_{\mathcal K}^c\) and therefore
> > > $$
> > > \Pr_{q\sim P_{\mathcal K}}(s(q)>\delta) \le \Pr(A_{\mathcal K}^c) \le \alpha
> > > $$
> > > For unknown queries, define $A_{\mathcal U} := \{ m_{\mathcal S}(q')\ge \Delta_{\mathcal S},\ m_{\mu}(q')\ge \Delta_\mu\}.$
> > >
> > > By assumption $(2)$ , $\Pr(A_{\mathcal U})\ge 1-\beta. \tag{26}$ On $A_{\mathcal U}$, for every $i\le T_{\mathrm{seen}}$, $d(q',\mathcal S_i)\ge \Delta_{\mathcal S}$ and $ \lVert q'-\mu_i\rVert_2\ge \Delta_\mu.$
> > >
> > > Thus, for every $i$, $D_i(q') = d(q',\mathcal S_i)^2+\lambda\lVert q'-\mu_i\rVert_2^2 \ge \Delta_{\mathcal S}^2+\lambda\Delta_\mu^2. $
> > >
> > > Taking the minimum over $i$, we obtain $s(q') = \min_i D_i(q') \ge \Delta_{\mathcal S}^2+\lambda\Delta_\mu^2 > \delta$,
> > >
> > > where the last inequality again follows from assumption $(3)$. Hence, the event $\{s(q')\le \delta\}$ can only occur if $A_{\mathcal U}^c$ occurs. Therefore
> > > $$
> > > \Pr_{q'\sim P_{\mathcal U}}(s(q')\le \delta) \le \Pr(A_{\mathcal U}^c) \le \beta.
> > > $$
> > > This completes the proof.

---

### Official Review · Reviewer_jdGN · 2026-03-11

**Soundness:** 3
**Presentation:** 2
**Significance:** 3
**Originality:** 3
**Overall Recommendation:** 4
**Confidence:** 3

**Summary:**

This paper investigates Open-Set Graph Class-Incremental Learning, aiming to address two core issues simultaneously during graph continuous learning: firstly, the model needs to learn progressively across multiple tasks without excessively forgetting previous knowledge; secondly, during testing, it needs to identify samples from future tasks or unknown semantic spaces, rather than forcibly classifying them into existing categories. To address this, the authors propose a framework centered on subspace-aware feature reshaping. Each learned task is represented as a task fingerprint composed of a centroid and a low-rank subspace, and during inference, a unified energy score is used to simultaneously handle task routing and the rejection of unknown samples.

**Compliance With Llm Reviewing Policy:**

Affirmed.

**Key Questions For Authors:**

Please see my weakness part

**Limitations:**

Yes

**Strengths And Weaknesses:**

Strengths

- This paper studies open-set graph class-incremental learning. This problem setting combines continual learning and unknown rejection, making it closer to real-world open environments than traditional closed-set incremental graph learning, thus attracting broad interest.

- The methodology is relatively clear. The authors represent each task as a fingerprint composed of a centroid and a low-rank subspace, and simultaneously achieve task routing and unknown rejection through a unified energy score. This framework is conceptually complete.

- The theoretical analysis and method design are largely consistent, providing a geometric interpretation of subspace-based routing and rejection.

- Experimental results show extremely strong performance in open-set detection and forgetting control on multiple datasets, with overall empirical performance being highly competitive.

Weaknesses

- My main concern with this paper is that the experimental results are overly idealistic. Several open-set metrics are close to perfect, and task routing remains 100% even under noise. This makes me want to see more sanity checks to further confirm the credibility of the results.

- Whether the current benchmark and protocol are challenging enough remains to be discussed; given the near-perfect results, I would prefer to see more difficult unknown constructions, more heterogeneous data settings, and more thorough robustness analysis.

- The theoretical part is generally more explanatory than a particularly strong foundational contribution, and therefore insufficient to be the primary source of the paper's persuasiveness.

---

> ### Author Rebuttal · Authors · 2026-03-30
>
> Thank you for your thoughtful and valuable feedback.
>
> **Weakness #1:**
>
> Our strong performance mainly comes from this paradigm being better suited to the open-set setting. TPP has already shown that separating task discrimination from task-specific prediction is clearly better than direct unified classification. Building on this, SAFER further isolates task subspaces with geometric fingerprints and unifies routing and rejection with a unified residual-energy criterion, thereby further amplifying this advantage.
>
>  As shown below, we have extended the experiments to include tasks with 3 and 5 classes.
>
> **Table: Performance with 5 classes per task.**
>
> | Dataset | CoraFull |         |          |          | Arxiv    |         |           |          | Product  |         |           |          |
> | :------ | :------- | :------ | :------- | :------- | :------- | :------ | :-------- | :------- | -------- | ------- | --------- | -------- |
> | Method  | AA↑      | AF↑     | AUC↑     | OSCR↑    | AA↑      | AF↑     | AUC↑      | OSCR↑    | AA↑      | AF↑     | AUC↑      | OSCR↑    |
> | DMSG    | 75.8±0.3 | 3.4±0.8 | 65.6±0.2 | 60.4±0.2 | 45.5±0.4 | 4.2±0.5 | 53.7±0.7  | 35.8±1.2 | 49.5±0.9 | 3.5±1.7 | 64.5±1.1  | 35.7±1.6 |
> | TPP     | 86.4±0.6 | 0.0±0.0 | 76.1±2.7 | 71.4±2.8 | 62.8±0.4 | 0.0±0.0 | 65.0±1.1  | 58.5±0.9 | 70.3±0.3 | 0.0±0.0 | 61.2±1.0  | 58.2±0.9 |
> | SAFER   | 90.9±0.1 | 0.0±0.0 | 99.5±0.5 | 91.2±0.2 | 82.3±0.2 | 0.0±0.0 | 100.0±0.0 | 81.8±0.2 | 79.7±0.1 | 0.0±0.0 | 100.0±0.0 | 93.1±0.1 |
>
> **Table: Performance with 3 classes per task.**
>
> Please see: https://anonymous.4open.science/r/SAFER-CB38/
>
> **Weakness #2:**
>
> For the benchmark and protocol, we adopt the widely used GCIL benchmark setting from CGLB to ensure a fair comparison with existing methods. We also include additional unknown-setting experiments to further examine robustness.
>
> - Tasks are ordered by similarity to evaluate discrimination under highly similar tasks.
>
> - Tasks are ordered in ascending order of data size to evaluate stability under different task scales.
>
> -  The last two tasks are treated as permanently unknown classes to evaluate rejection under a stricter open-set setting.
>
> **Table: Performance in similarity order.**
>
> | Dataset | CoraFull |         |          |          | Arxiv    |          |           |          | Product  |           |           |          |
> | :------ | :------- | :------ | :------- | :------- | :------- | :------- | :-------- | :------- | -------- | --------- | --------- | -------- |
> | Method  | AA↑      | AF↑     | AUC↑     | OSCR↑    | AA↑      | AF↑      | AUC↑      | OSCR↑    | AA↑      | AF↑       | AUC↑      | OSCR↑    |
> | DMSG    | 77.8±0.4 | 1.6±0.6 | 66.2±0.3 | 62.4±0.1 | 47.6±0.6 | -5.4±0.4 | 53.7±0.2  | 37.4±0.5 | 54.0±1.6 | -14.2±4.3 | 61.7±0.5  | 36.9±1.2 |
> | TPP     | 91.6±0.0 | 0.0±0.0 | 80.9±0.1 | 79.9±0.0 | 85.4±0.1 | 0.0±0.0  | 58.8±0.1  | 57.7±0.1 | 87.2±0.0 | 0.0±0.0   | 69.1±0.0  | 68.5±0.0 |
> | SAFER   | 93.5±0.2 | 0.0±0.0 | 97.0±0.1 | 93.5±0.0 | 90.0±0.0 | 0.0±0.0  | 100.0±0.0 | 91.4±0.4 | 94.6±0.0 | 0.0±0.0   | 100.0±0.0 | 97.8±0.0 |
>
> **Table: Performance in ascending order and under Permanent Unknown-Task Setting**
>
> Please see:  https://anonymous.4open.science/r/SAFER-CB38/
>
> **Weakness #3:**
>
> Our theoretical goal is not to establish a foundational theory for the CGL, but to provide formal support for the unified decision criterion in this paper. The three propositions mentioned are intended to theoretically justify the rationality of our unified residual-energy criterion:  Proposition 3.1 corresponds to the distance from samples to the task affine subspace; Propositions 3.2 and 3.3 further show that if task fingerprints are sufficiently distinguishable in the representation space, this unified energy naturally provides a margin for unknown rejection and task routing.
>
> The modules mentioned in Section 3.5 are designed precisely around the above geometric conditions. We use frozen anchors and prompt components as well as adapter, PPR  and so on to compress the intra-class radius and suppress cross-task drift, thereby promoting the geometric conditions required by the propositions.
>
> In ablation study, The results in Fig. 7 show that the full model learns more compact intra-class and better separated inter-class representations, supporting the propositions. In addition, after removing the router, unknown detection degrades significantly, indicating that these geometric improvements are translated into a more reliable routing/rejection margin.
>
> We hope these additional analyses and clarifications address your concerns more comprehensively.
>
> [1] Replay-and-Forget-Free Graph Class-Incremental Learning: A Task Proﬁling and Prompting Approach. NeurIPS 2024
>
> [2] CGLB: Benchmark Tasks for Continual Graph Learning. NeurIPS 2022

---

> > ### Author Rebuttal · Reviewer_jdGN · 2026-04-01
> >
> > Thank you for the detailed rebuttal and clarifications. My questions have been largely addressed. I am inclined to maintain my positive score.

---

> > > ### Author Response · Authors · 2026-04-03
> > >
> > > We sincerely appreciate your recognition of our work and the time and effort you have devoted as a reviewer!

---

### Decision · Program_Chairs · 2026-04-30

**Decision:**

Accept (regular)

**Comment:**

The paper studies open-set graph class-incremental learning, a practical and challenging setting. Its strengths include a well-motivated problem formulation, a clear and logically designed methodology using subspace-aware feature reshaping with a unified energy score, strong empirical performance across multiple benchmarks, and theoretical support for the decision mechanism. The main weaknesses noted were concerns about the overly idealistic experimental results, a lack of recent baselines, and the theoretical analysis being somewhat detached from the method. After the rebuttal, most concerns have been largely addressed, while one reviewer still finds the theoretical integration insufficient. Overall, the paper makes a solid contribution to an important problem.